# Strong anthropogenic control of secondary organic aerosol formation from isoprene in Beijing

Daniel J. Bryant[1], William J. Dixon[1], James R. Hopkins[1,2], Rachel E. Dunmore[1], Kelly L. Pereira[1], Marvin Shaw[1,2], Freya A. Squires[1], Thomas J. Bannan[3], Archit Mehra[3], Stephen D. Worrall[3±], Asan Bacak[3β], Hugh Coe[3], Carl J. Percival[3φ], Lisa K. Whalley[4,5], Dwayne E. Heard[4], Eloise J. Slater[4], Bin Ouyang[6,7], Tianqu Cui[8ω], Jason D. Surratt[8], Di Liu[9υ], Zongbo Shi[9,10], Roy Harrison[9], Yele Sun[11], Weiqi Xu[11], Alastair C. Lewis[1,2], James D. Lee[1,2], Andrew R. Rickard[1,2], Jacqueline F. Hamilton[1]

[1] Wolfson Atmospheric Chemistry Laboratories, Department of Chemistry, University of York, York, UK
[2] National Centre for Atmospheric Science, University of York, York, UK
[3] School of Earth and Environmental Sciences, The University of Manchester, Manchester, UK
[4] School of Chemistry, University of Leeds, Leeds, UK
[5] National Centre for Atmospheric Science, University of Leeds, Leeds, United Kingdom
[6] Lancaster Environment Centre, Lancaster University, Lancaster, UK
[7] Department of Chemistry, University of Cambridge, Cambridge, UK
[8] Department of Environmental Sciences and Engineering, Gillings School of Global Health, University of North Carolina, Chapel Hill, USA
[9] School of Geography Earth and Environmental Sciences, the University of Birmingham, Birmingham, UK
[10] Institute of Surface-Earth System Science, Tianjin University, Tianjin, China
[11] Institute of Atmospheric Physics, Chinese Academy of Sciences, Beijing, People's Republic of China

± Now at Chemical Engineering and Applied Chemistry, School of Engineering and Applied Science, Aston University, Birmingham, UK
β Turkish Accelerator & Radiation Laboratory, Ankara University Institute of Accelerator Technologies, Ankara, Turkey
φ Now at Jet Propulsion Laboratory, California Institute of Technology, 4800 Oak Grove Drive, Pasadena, CA, USA
ω Now at Laboratory of Atmospheric Chemistry, Paul Scherrer Institute, 5232 Villigen, Switzerland
υ Now at State Key Laboratory of Organic Geochemistry and Guangdong Provincial Key Laboratory of Environmental Protection and Resources Utilization, Guangzhou Institute of Geochemistry, Chinese Academy of Sciences, Guangzhou 510640, China

*Correspondence to: Jacqueline Hamilton (jacqui.hamilton@york.ac.uk)*

**Abstract:** Isoprene-derived secondary organic aerosol (iSOA) is a significant contributor to organic carbon (OC) in some forested regions, such as tropical rainforests and the Southeast US. However, its contribution to organic aerosol in urban areas, with high levels of anthropogenic pollutants, is poorly understood. In this study we examined the formation of anthropogenic-influenced iSOA during summer in Beijing, China. Local isoprene emissions and high levels of anthropogenic pollutants, in particular $NO_X$ and particulate $SO_4^{2-}$, led to the formation of iSOA under both high- and low-NO oxidation conditions, with significant heterogeneous transformations of isoprene-derived oxidation products to particulate organosulfates (OSs) and nitrooxy-organosulfates (NOSs). Ultra-high performance liquid chromatography coupled to high-resolution mass spectrometry was combined with a rapid automated data processing technique to quantify 31 proposed iSOA tracers in offline $PM_{2.5}$ filter extracts. The co-elution of the inorganic ions in the extracts caused matrix effects that impacted two authentic standards differently. The average concentration of iSOA OSs and NOSs was 82.5 ng m$^{-3}$, around three times

higher than the observed concentrations of their oxygenated precursors (2-methyltetrols and 2-methylglyceric acid). OS formation was dependant on both photochemistry and sulfate available for reactive uptake as shown by a strong correlation with the product of ozone ($O_3$) and particulate sulfate ($SO_4^{2-}$). A greater proportion of high-NO OS products were observed in Beijing compared to previous studies in less polluted environments. The iSOA derived OSs and NOSs represented on average 0.62 % of the oxidised organic aerosol measured by aerosol mass spectrometry, but this increased to ~3 % on certain days. These results indicate for the first time that iSOA formation in urban Beijing is strongly controlled by anthropogenic emissions and results in extensive conversion to OS products from heterogenous reactions.

**1 Introduction**

Rapidly developing countries such as China often experience very poor air quality. Beijing regularly experiences periods of very high particle pollution, with annual and 24-hourly levels well above World Health Organisation guidelines (Chan et al., 2008; Hu et al., 2014). Premature mortality, as a result of respiratory illness, cardiovascular disease and cancer, has been associated with exposure to poor air quality (Dockery et al., 1993; Pope et al., 2000, 2006; Jerrett et al., 2009; Beelen et al., 2014; Laurent et al., 2014; Ostro et al., 2015). Lelieveld et al. (2015) estimated that 1.36 million premature deaths in China in 2010 were a result of exposure to outdoor air pollution. By far the most dangerous pollutant to health in China are particles less than 2.5 microns in diameter, known as $PM_{2.5}$, with a recent study suggesting that a 50 % reduction in excess mortality requires a 62 % reduction in $PM_{2.5}$ in the Beijing-Tianjin-Hebei (BTH) region (Hu et al., 2017a).

Previous measurements using aerosol mass spectrometry (AMS) indicate that $PM_1$ in Beijing is mainly composed of sulfate, nitrate, ammonium and organics (Hu et al., 2016. Zhang et al., 2013). Positive Matrix Factorisation of AMS measurements indicate that oxidised or secondary organic aerosol (SOA) can make up a substantial fraction of the $PM_1$ mass (> 25 %), even in urban areas, but the sources of this material are still poorly understood (Zhang et al., 2011; Sun et al., 2018). Hu et al. (2017a) estimated that exposure to SOA was responsible for 0.14 million deaths in China in 2013 based on mass contribution alone, ranging from < 1 % to 23 % source contributions to $PM_{2.5}$ depending on location. Zhang et al. (2017) used [14]C measurements to determine that non-fossil emissions are generally a dominant contributor of secondary organic carbon (SOC) in Beijing, with a larger contribution in summer as a result of increased biogenic volatile organic compounds (VOCs) emissions.

Hu et al. (2017b) updated the Community Multi-scale Air Quality (CMAQ) model with updated SOA yields and a more detailed description of SOA formation from isoprene oxidation. Removing all anthropogenic pollutants from the model resulted in a huge drop in isoprene SOA concentrations, indicating that controlling anthropogenic emissions would result in reduction of both anthropogenic and biogenic SOA. The predicted SOA was dominated by isoprene in summer across China and in four cities (Beijing, Guangzhou, Shanghai, Chengdu) with concentrations up to 30 µg m$^{-3}$ in Beijing. However, there is currently very little observational evidence to support such high SOA mass concentrations from

isoprene oxidation in these Chinese cities. The widely used SOA tracer method (Kleindienst et al., 2007) has been used extensively to estimate the fraction of isoprene-derived SOA (iSOA) across China. Ding et al. (2014) studied SOA at 14 Chinese sites and found that iSOA dominated the apportioned SOA mass (46 ± 14 %), with it contributing between 0.4 – 2.17 µg m$^{-3}$ and an average of 1.59 µg m$^{-3}$ in Beijing. However, only a very limited subset of VOC precursors was included, and this method fails to account for heterogeneous formation processes. To overcome some of these limitations, Wang et al. (2017) used tracer-based source apportionment of PM$_{2.5}$ with positive matrix factorisation in the Pearl River Delta region during summer. They identified an iSOA factor that contributed up to 4 µg m$^{-3}$ in Guangzhou, and up to 11 % of the total SOC.

A multitude of studies have examined iSOA formation (Pandis et al., 1991; Edney et al., 2005; Kroll et al., 2006; Dommen et al., 2006; Kleindienst et al., 2006; Ng et al., 2006; Surratt et al., 2007a, 2007b, 2008, 2010; Ng et al., 2008; Paulot et al., 2009; Chan et al., 2010; Chhabra et al., 2010; Nguyen et al., 2011, 2014, 2015; Zhang et al., 2011, 2012; Lin et al., 2013; Xu et al., 2014; Krechmer et al., 2015; Clark et al., 2016; Riva et al., 2016a, 2016b); however, the magnitude of iSOA formed can be vastly different from study to study (Carlton et al., 2009). Furthermore, there have been limited field measurements to establish if these estimates are representative of urban environments (Wang et al., 2018; Li et al., 2018; Glasius et al., 2018; Le Breton et al., 2018; Hettiyadura et al., 2018,2019; Rattanavaraha et al., 2016; Budisulistiorini et al., 2013).

iSOA formation during the daytime is dominated by reaction with hydroxyl radicals (OH), with the concentrations of NO having a strong influence on the reaction products (Wennberg et al., 2018 and references therein). Under low-NO conditions the isoprene peroxyradicals (RO$_2$) can react with hydroperoxy radicals (HO$_2$) to form isoprene hydroxyhydroperoxides (ISOPOOH). The ISOPOOH isomers can react further with OH to form isoprene epoxydiol isomers (β− or δ-IEPOX) (Paulot et al., 2009), which can undergo uptake into acidified sulfate particles to form 2-methyltetrol organosulfates (2-MT-OS) (Surratt et al., 2010. Lin et al., 2012). Under high-NO conditions, isoprene RO$_2$ can react with NO to form alkoxy radicals (RO) producing methacrolein (MACR) and methyl vinyl ketone (MVK) as the main reaction products. The reaction of MACR with OH, and subsequent addition of NO$_2$, leads to the methylacrloylperoxynitrate (MPAN), which reacts with OH to produce hydroxymethylmethyl-α-lactone (HMML) (Nguyen et al., 2015) or methyacrylic epoxide (MAE) (Lin et al., 2013). HMML is thought to be the more abundant product compared to MAE (Nguyen et al., 2015). Subsequent uptake of HMML into wet sulfate aerosols is proposed to lead to either 2-methylglyceric acid (2-MG) or its organosulfate derivative (2-MG-OS), as well as their dimers and higher order oligomers (Surratt et al., 2006; Surratt et al., 2010; Lin et al., 2013; Nguyen et al., 2015). Recently Schwantes et al. (2019) showed that under ambient conditions the formation of SOA from low-volatility nitrates and dinitrates, formed via reactions of isoprene derived RO$_2$ with NO, is also important. Chamber-derived SOA yields for OH chemistry are variable depending on the experimental conditions, but are generally low (<10 %). The addition of acidified sulfate aerosol, accounting for wall losses and using more atmospherically relevant radical lifetimes can lead to significantly higher SOA yields in chamber studies (Surratt et al., 2010; Lin

et al. 2012; Gaston et al., 2014). However, recent work has revealed that isoprene SOA formation can be suppressed when viscous organic coatings are present on acidified sulfate aerosol, impeding the multiphase chemistry of IEPOX yielding additional SOA (Riva et al., 2016b; Zhang et al., 2018; Riva et al., 2019).

Observations using aerosol mass spectrometry (AMS) indicate that IEPOX-derived SOA can make up a significant fraction of organic aerosol in isoprene-rich environments, such as Borneo (23 %; Robinson et al., 2011), the Amazon (34 %; Chen et al., 2015) and the South East US (33-40 %; Budisulistiorini et al., 2013; Budisulistiorini et al., 2016; Rattanvaraha et al., 2017). Hu et al. (2015) compared previous AMS studies and found a magnitude lower average IEPOX-SOA signal in urban studies ($f_{C5H6O}$ = 0.17 %) compared to those in isoprene-rich regions ($f_{C5H6O}$ = 2.2 %). The average IEPOX-SOA concentration measured in Nanjing, a polluted city in Eastern China, in August 2013 was 0.33 μg m$^{-3}$ (Zhang et al., 2017). This represented only 3.8 % of the total OA, indicating there is limited formation of IEPOX under high-NO$_x$ conditions (average NO$_x$ = 21 ppb). He et al. (2018) found higher concentrations of the low-NO isoprene SOA tracers (average = 121 ng m$^{-3}$) than the high-NO iSOA tracers (average = 9 ng m$^{-3}$) at a regional background site (Wanqingsha) situated within the heavily polluted Pearl River Delta Region. Only two high-NO iSOA tracers were measured (2-MG and 2-MG-OS), which could lead to a significant underestimate of the strength of the high NO$_x$ pathway. Wang et al. (2018) measured a range of OSs at a regional site 38 km north east of Beijing during May-June 2016. Isoprene-derived OSs ranged from 0.9-20 ng m$^{-3}$, with a mean isoprene-derived OS concentration of 14.8 ng m$^{-3}$. In both these studies, the ratio of the average concentration of the commonly used OS tracers from the low NO versus the high NO pathways was close to 1.5 (2-MT-OS:2-MG-OS; Beijing = 1.47, Wanqingsha = 1.57) indicating that even in polluted environments low-NO oxidation chemistry can play a significant role in iSOA formation.

SOA formed from anthropogenic and other biogenic (monoterpenes and sesquiterpenes) sources have also been studied. Thousands of organic species including hundreds of OS and nitroxy OS (NOS) species have been identified studies from a range of precursors using UHPLC/ESI-HR-MS from ambient aerosol samples (Wang et al., 2016, Wang et al., 2017). Brüggemann et al. (2019) quantified with authentic standards both monoterpene OS (MT-OS) and sesquiterpene OS (SQT-OS) species in Melpitz, Germany and Wangdu, China. They found median daytime concentrations for Melpitz and Wangdu for 52 MT-OS species of 12.15 ng m$^{-3}$ and 38.19 ng m$^{-3}$ respectively. For the 5 SQT-OS species, median concentrations were 0.3 ng m$^{-3}$ and 3.90 ng m$^{-3}$ for daytime concentrations respectively, much lower than the iSOA OS species quantified in this study. Riva et al. (2016) identified OS species from the photo-oxidation of $C_{10}$ – $C_{12}$ alkanes, which were then characterised in ambient aerosol samples collected in Lahore, Pakistan and Pasadena, CA, USA. High concentrations of OS species were identified in Lahore, with the largest observed concentration arising from a cycodecane OS species ($C_{10}H_{16}O_7S$) with a concentration of 35.93 ng m$^{-3}$.

The lack of molecular-level measurements of iSOA in highly polluted urban areas makes it difficult to determine the role of isoprene in summer haze episodes in Beijing. To investigate the formation of iSOA in Beijing, offline $PM_{2.5}$ filter samples were collected during summer 2017 as part of the Atmospheric Pollution and Human Health program (Shi et al., 2019). The filters were extracted and then screened using a sensitive and selective high throughput method based on ultra-high-performance liquid chromatography coupled to ultra-high-resolution mass spectrometry equipped with electrospray ionization (UHPLC/ESI-HR-MS). High-time resolution filter sampling allowed the formation and evolution of iSOA to be studied, with observed concentrations strongly controlled by levels of anthropogenic pollutants.

## 2 Experimental

### 2.1 PM$_{2.5}$ filter sampling and extraction

Aerosol samples were collected between the 18[th] May and 24[th] June 2017 at the Institute of Atmospheric Physics (IAP) in Beijing, China. This sampling was part of the Sources and Emissions of Air Pollutants in Beijing (AIRPOLL-Beijing) project, as part of the wider Atmospheric Pollution and Human Health in a Chinese Megacity (APHH-Beijing) programme (Shi et al., 2019). $PM_{2.5}$ filter samples were collected using an ECOTECH HiVol 3000 (Ecotech, Australia) high-volume air sampler with a selective $PM_{2.5}$ inlet, with a flow rate of 1.33 $m^3$ $min^{-1}$. Filters were baked at 500 $^{o}C$ for five hours before use. After collection, samples were wrapped in foil, and then stored at -20 $^{o}C$ and shipped to the laboratory for offline analysis. Samples were collected at a height of 8 m, on top of a building in the IAP complex. Samples were collected every 3 hours during the day, approximately between 08:30 and 17:30 and then one sample was collected overnight between 17:30 and 08:30. Hourly samples were also taken on certain days towards the end of the sampling period on high pollution days. 24-hour samples were also collected using a Digitel high volume $PM_{2.5}$ sampler at the same location.

The extraction of the organic aerosol from the filter samples was based on the method in Hamilton et al., (2008). Initially, an 8[th] of the filter was cut up into roughly 1 $cm^2$ pieces and stored in a vial. 4 ml of LC-MS grade $H_2O$ was then added to the sample and left for two hours. The samples were then sonicated for 30 minutes. A small subset (3) of the filter samples were also extracted via orbital shaker and no appreciable difference was found in the concentrations of the iSOA tracers compared to sonication. Using a 2 ml syringe, the water extract is then pushed through a 0.22 μm filter (Millipore) into another sample vial. An additional 1 mL of water was added to the filter sample, then extracted through the filter, to give a combined aqueous extract. This extract was then reduced to dryness using a vacuum solvent evaporator (Biotage, Sweden). The dry sample was then reconstituted in 1 mL 50:50 $MeOH:H_2O$ solution for offline chemical analysis.

### 2.2 Ultra-high performance liquid chromatography tandem mass spectrometry (UHPLC-MS$^2$)

The water-soluble fraction of the filter samples were analysed using UHPLC-full scan-ddMS$^2$, using an Ultimate 3000 UHPLC (Thermo Scientific, USA) coupled to a Q-Exactive Orbitrap MS (Thermo Fisher Scientific, USA) with a heated electrospray ionisation (HESI). The UHPLC method uses a reverse phase

5 μm, 4.6 x 100mm, Accucore column (Thermo Scientific, UK) held at 40 °C. The mobile phase consists of LC-MS grade water and 100 % MeOH (Fisher Chemical, USA). The water was acidified using 0.1 % formic acid to improve peak resolution. The injection volume was 2 μL. The solvent gradient was held for a minute at 90:10 $H_2O$:MeOH, then changed linearly to 10:90 $H_2O$:MeOH over 9 minutes, then held for 2 minutes at this gradient before returning to 90:10 $H_2O$:MeOH over 2 minutes and then held at 90:10 for the remaining 2 minutes, with a flow rate of 300 μL min$^{-1}$. Due to the wide range of compounds studied, poor retention was observed for some species (RT < 0.8min). These species closely eluted to the dead time of the column where inorganic sulfate ions eluted (0.67 min). To check for ionisation artefacts, a aqueous solution containing 20 ppm ammonium sulfate, 1ppm 2-methytetrol and 1ppm 2-methylglyceric acid was ran under the same conditions as the filter samples to check for organosulfate formation (2-MT-OS and 2-MG-OS respectively). No MG-OS formation was observed and <0.5% conversion was seen for the 2-MT. This therefore rules out adduct formation for the two most important iSOA species, 2-MT-OS and 2-MG-OS, however due to the lack of authentic standards and the complexity of the samples, adduct formation throughout the entire chromatogram could still be occurring. At this stage, there is not enough evidence to say either way if adducts are forming or not. The mass spectrometer was operated in negative mode using full scan ddMS$^2$. The scan range was set between 50 - 750 m/z, with a resolution of 70,000. The ESI voltage was 4 kV, with capillary and auxiliary gas temperatures of 320 °C. The number of most abundant precursors for MS$^2$ fragmentation per scan was set to 10. The samples were run in batches of 70, in a repeating sequence of 5 samples followed by one blank, each filter sample was run only once. The calibrations were run separately after the samples were finished, in the following sequence; (3 X same concentration) X number of standards in calibration curve from the lowest concentration to the highest followed by 2 blanks. The quantification method will be discussed in the results section (3.3).

## 2.3 Construction of accurate mass library

A mass spectral library was built using the compound database function in TraceFinder 4.1 General Quan software (Thermo Fisher Scientific, USA). Each compound was input into the compound library in the generic form: $C_cH_hO_oN_nS_s$ (where c, h, o, n, and s represent the number of carbon, hydrogen, oxygen, nitrogen and sulfur atoms respectively). From literature, species were identified, searched for in the ambient samples according to their accurate mass, and then the retention time (RT) of each isomer was obtained. Using previously observed iSOA products from literature, extracted ion chromatograms were plotted for each *m/z* value from a small subset of ambient samples and the retention time (RT) of observed species/isomer were obtained. For most of the OS species in this study the separation was not good enough to see individual isomers and only one peak was observed, which was added to the library. For the NOS species, individual isomers could be resolved, and each isomer was added to the library based on its retention time. The accurate masses, RT and literature references for iSOA tracers are shown in Table ~~1~~2.

## 2.4 Automated method for SOA tracer analysis

The UHPLC/ESI-HR-MS data for each ambient sample and standard was analysed using TraceFinder[TM].
Tracefinder extracted the OS/NOS tracer peak areas from each ambient sample chromatogram using the
library based on RT and accurate mass. The mass tolerance of the method was set to 2 ppm and the
retention time window was set to 30 s, although for species with multiple isomers present, the integration
was checked to make sure the same peaks were not being integrated twice, and the window changed

accordingly. The peak tailing factor was set to 2.0 to reduce the integration of the peak tails. The
minimum signal to noise (S/N) for a positive identification was set to 3.0.  Using the output from
TraceFinder, an *in-house* R code script was developed to combine the identified species and peak areas
with the correct filter sampling date/time midpoint and volume of air sampled. Calibration curves from
the standards were then obtained (as discussed in section 3.3), and the intercept and gradient inputted to

quantify the iSOA tracer concentrations in the extract.  These quantified values were then converted to
the mass on the whole filter and divided by the volume of air sampled for that filter sampling period and
converted to units of ng m$^{-3}$. Higher time resolution data were averaged to the filter sampling times. It
should be noted that MS$^2$ was used to check that the iSOA species fragmented to give typical OS fragment
ions.


## 2.5 Hydrophilic Liquid Interaction Chromatography (HILIC).

A subset of filters (n=15) were also analysed at the University of North Carolina (UNC) using a newly
developed HILIC method interfaced to high-resolution quadrupole time-of-flight mass spectrometry
equipped with ESI (i.e., HILIC/ESI-HR-QTOFMS) (Cui et al., 2018). Briefly, filters were extracted with

22 mL of LC/MS-grade methanol by 45 min of sonication; the samples were first extracted for 23 min,
the water bath replaced with cool water, and then extracted again for 22 min.  This was done to make
sure the water bath contained within the sonicator did not reach above 30 C. Extracts were filtered
through polypropylene membrane syringe filters in order to remove insoluble filter fibres and soot
particles. The extracts were dried under a gentle stream of nitrogen gas.  Dried methanol extracts were

reconstituted with 150 μL of 95:5 (v/v) LC/MS-grade acetonitrile/Milli-Q water. Operating details of the
HILIC/ESI-HR-QTOFMS used for these samples is also summarized by Cui et al. (2018).

## 2.6 Gas Chromatography – Mass Spectrometer

Details of the measurement procedure used can be found elsewhere (Fu et al., 2010). Briefly, filter

samples were extracted with dichloromethane/methanol (2:1 v/v), filtered through quartz wool packed
in a Pasteur pipette, concentrated using a rotary evaporator under vacuum, and blown down to dryness
with pure nitrogen gas. The extracts were derivatized and diluted with n-hexane containing the internal
standard prior to GC-MS analysis. Separation was performed on a fused silica capillary column (DB-
5MS: 30 m × 0.25 mm × 0.25 μm). The MS detection was conducted in electron ionization (EI) mode at

70 eV, scanning from 50 to 650 Da. Individual compounds were identified by comparison of mass spectra
with those of authentic standards or literature data. 2-methylglyceric acid, C$_5$-alkene triols (the sum of
*cis*-2-methyl-1,3,4-trihydroxy-1-butene, *trans*-2-methyl-1,3,4-trihydroxy-1-butene, and 3-methyl-2,3,4-
trihydroxy-1-butene), and 2-methyltetrols (the sum of 2-methylthreitol and 2-methylerythritol) were

quantified using the response factor of *meso*-erythritol. Field blank filters were treated as the real samples for quality assurance. Target compounds were not detected in the blanks.

### 2.7 High-Resolution Aerosol Mass Spectrometry measurements

The size-resolved non-refractory submicron aerosol species at the same site were measured by an Aerodyne high-resolution time-of-flight aerosol mass spectrometer (HR-ToF-AMS) at a time resolution of 5 min. The elemental ratios of hydrogen-to-carbon (H:C) and oxygen-to-carbon (O:C) of OA were determined, and the sources of OA were analysed with positive matrix factorisation. Six OA factors were identified in summer including two primary factors; hydrocarbon like OA (HOA), cooking OA (COA), and three oxidised OA factors with increasing degrees of oxidation, OOA1 (O:C = 0.53), OOA2 (O:C = 0.74), OOA3 (O:C = 1.18).

### 2.8 Iodide CIMS

A time of flight chemical ionisation mass spectrometer (ToF-CIMS) (Lee et al. 2014; Priestley et al. 2018) using an iodide ionisation system coupled with a filter inlet for gases and aerosols (FIGAERO) was deployed here to make near simultaneous, real-time measurements of both the gas- and particle-phase chemical composition. The instrument was originally developed by Lopez-Hilfiker et al. (2014) and is described and characterised in more detail by Bannan et al., (2019). The experimental set up employed by the University of Manchester ToF-CIMS is described in Zhou et al., (2019). Only gas phase data is presented herein.

Field calibrations were regularly carried out using known concentration of formic acid in gas mixtures made in a custom-made gas phase manifold. A range of other species were calibrated for after the campaign, and relative calibration factors were derived using the measured formic acid sensitivity during the in-situ calibrations (Bannan et al. 2015). Offline calibrations after the field work campaign were performed specific to the isoprene oxidation species observed here. IEPOX ($C_5H_{10}O_3$) synthesized by the University of North Carolina, Department of Environmental Sciences & Engineering was specifically calibrated for. Known concentrations were deposited on the FIGAERO filter in various amounts and thermally desorbed using a known continuous flow of nitrogen over the filter. For the isoprene nitrate; $C_5H_9NO_4$ there was no direct calibration source available and concentrations using the calibration factor of $C_5H_{10}O_3$ are presented here.

### 2.9 Gas-phase measurements

Additional gas-phase measurements were collected at the site from an elevated inlet at 8 m. Data included Nitrogen oxide, NO, measured by chemiluminescence with a Thermo Scientific Model 42i $NO_x$ analyser and Nitrogen dioxide, $NO_2$, was measured using a Teledyne Model T500U Cavity Attenuated Phase Shift (CAPS) spectrometer. The sum of the $NO_y$ species was measured using a Thermo Scientific Model 42C $NO_x$ analyser and a heated molybdenum converter at the sample inlet. The molybdenum converter reduces $NO_y$ compounds to NO allowing measurement by chemiluminescence. Ozone, $O_3$, was measured

using a Thermo Scientific Model 49i UV photometric analyser. All instruments were calibrated throughout the measurement period, with a 'zero' or 'background' calibration using a Sofnofil/charcoal trap. Span (high concentration) calibrations were carried out using gas standards. Both the Thermo Scientific 42i and 42C instrument calibrations are traceable to the National Physical Laboratories (NPL) NO scale. The meteorological variables of wind speed, wind direction, relative humidity (RH), and temperature were measured at 102 m on the IAP 325 m meteorological tower.

Observations of VOCs were made using a dual-channel GC with flame ionisation detectors (DC-GC-FID). Air was sampled at 30 L min$^{-1}$ at a height of 5m, through a stainless-steel manifold (½" internal diameter). 500 mL subsamples were taken, dried using a glass condensation finger held at -40ºC and then pre-concentrated using a Markes Unity2 pre-concentrator on a multi-bed Ozone Precursor adsorbent trap (Markes International Ltd). These samples were then transferred to the GC over for analysis following methods described by Hopkins et al. (2011).

Further details of the following additional gas phase instrumentation can be found in the SI and Shi et al., 2019. Isoprene was also measured at a height of ~102 m using a Voice200 Selected ion flow tube mass spectrometer (SIFT-MS, Syft Technologies, Christchurch, New Zealand). OH, HO$_2$ and RO$_2$ concentrations were measured using Fluorescence Assay by Gas Expansion (FAGE) (Whalley et al., 2010) and NO$_3$ concentrations were measured using Broadband cavity enhanced absorption spectrometry (Le Breton et al., 2014).

## 3 Results and discussion

The field campaign was conducted at the Institute of Physics, Beijing, situated between the third and fourth ring roads (Shi et al., 2019). The site is typical of central Beijing, surrounded by residential and commercial properties and is near several busy roads. It is also close to several green spaces including a tree-lined canal to the south and the Olympic forest park to the north-east, providing sources for local isoprene emissions.

### 3. 1 Isoprene gas phase concentrations and loss processes

Isoprene was measured hourly using the DC-GC-FID between 18/05/2017 – 20/06/2017 and the observed concentrations are shown in Figure 1, alongside NO, NO$_2$ and ozone. The mean mixing ratio of isoprene was 0.53 ppb, with a maximum of 2.9 ppb on the 16/06/2017. The ambient temperature ranged from 16 to 38 °C. Day-time isoprene mixing ratios increased with temperature, with all isoprene mixing ratios above 1 ppb occurring when the temperature was > 25 °C. The average diurnal profile of isoprene in Figure 3a shows low values overnight (< 50 ppt), with a rapid increase at 6 am reaching a maximum of around 1 ppb by the afternoon. The mixing ratio rapidly decreased after 18:00 and returned to very low values by around 22:00. There was strong a correlation between the isoprene mixing ratio measured at 8 m by the DC-GC and at 102 m using the SIFT-MS ($R^2$ = 0.77). The SIFT-MS measurements were therefore used to investigate the correlation with iSOA tracers when no DC-GC data

was available. The slope of the linear fit between the two data sets was 0.67, indicating a loss of around 30% of the isoprene during transport from the ground to the tower (100m).

Using the average observed diurnal profiles of the main atmospheric oxidants, OH, ozone and $NO_3$ (shown in SI Figure S1), and isoprene (Figure 3a), the isoprene loss rate was calculated (rate of loss = $k_{ox}$[Oxidant][Isoprene]) and is shown in Figure 4a. The IUPAC rate constants that were used in the calculation for $NO_3$, $O_3$ and OH were $7 \times 10^{-13}$, $1.27 \times 10^{17}$, $1 \times 10^{10}$ cm$^3$ molecule$^{-1}$ s$^{-1}$ respectively. The percentage contribution of each oxidant to the average diurnal isoprene loss rate is shown in Figure 4b.

During the day, OH is responsible for over 90 % of isoprene loss, with $NO_3$ becoming relatively more important from 18:00 until around 03:00, although the amount of isoprene available to react rapidly decreased during this time period. OH chemistry is still an important loss route at night (>30 %) owing to night-time OH sources, such as the ozonolysis of alkenes (Lu et al., 2014). Loss of isoprene via ozonolysis however is a minor route, contributing <15 %. During the daytime (10:00-15:00), the lifetime

of isoprene was on average around 20 minutes, increasing to a maximum of around 6 hours at 03:00. While the high levels of oxidants lead to a short isoprene lifetime during the day, the ambient concentrations of isoprene are still maintained at the ppb level. This indicates that there are significant local emissions of isoprene impacting the measurement site and therefore a high potential for the formation of iSOA in this urban environment.


**3.2 Anthropogenic tracers**

A range of gas phase anthropogenic tracers were measured during the campaign as discussed in Shi et al., 2019. Figure 1 shows the time series of NO, $NO_2$, $O_3$ and particulate sulfate during the part of the campaign analysed in this study. Table 1 shows the average, maximum and minimum concentrations for

these anthropogenic pollutants. NO mixing ratios ranged from less than 10.1 ppbv to 104 ppbv, and a mean concentration during the filter sampling period of 5.1 ppb. The highest concentrations generally occurred in the morning 04:00-07:00 and steadily decreased during the day. On some days, the mixing ratio of NO was very low in the afternoon, as a result of reaction with ozone and other unknown sinks (Newland et al., 2020). The mean mixing ratio of $NO_2$ was 22.3 ppbV, much higher than NO, with a

range of 3.7 to 95 ppbv. $NO_2$ peaked between 06:00-07:00 and decreased to a minima at 14:00 and then steady increased until about 20:00. High afternoon concentrations of $O_3$ (>80 ppb) were found on most days, with a maximum observed mixing ratio of 182 ppbv. Night time $O_3$ levels were much lower due to reduced photochemistry and reaction with NO, although on some nights $O_3$ levels were maintained above 40 ppbv, as shown in Figure 1. particulate sulfate concentrations, measured by AMS are also

shown in figure 1. Sulfate ranged from 0.7 to 21.7 µg m$^{-3}$, with an average of 5.5 µg m$^{-3}$. The time series shows a number of periods of high sulfate concentrations and these generally matched periods of increased PM$_{2.5}$ (see figure 9). Figure 2 shows the wind direction dependent concentrations of particulate sulfate for the sampling period in a pollutionRose plot (R, Openair package). There is a strong source of sulfate from the south of the sampling site, which is enhanced under the highest wind speeds. Previous

studies have shown a strong source of pollution from the south west of Beijing, which is where many industrial factories are located (Wang et al., 2005).

 **3.3 Isoprene SOA in Beijing**

Using the high throughput screening method described, the peak areas of 31 potential isoprene-derived OSs and NOSs, which are known iSOA tracers, were measured in 132 PM$_{2.5}$ filter extracts. The full list of iSOA tracers, along with their measured *m/z* and molecular formula is shown in Table 2, ordered by descending average concentration (weighted by filter sampling time and reported in ng m$^{-3}$) during the

campaign. The isoprene SOA tracers identified in this study are correlated towards themselves as well as common anthropogenic tracers in a correlation plot (R, Openair, CorPlot), shown in Figure 5. The correlation plot highlights the correlations of the iSOA tracers to each other as well as the moderate to strong correlations towards some of the anthropogenic pollutants as discussed in further sections.

**3.4 Quantification of isoprene OS tracers**

Initially, two synthesised isoprene-derived OS standards (2-MT-OS and 2-MG-OS, Cui et al., 2018; Rattanavaraha et al., 2016) were used to produce calibration curves. Both standards gave strong linear calibration curves (R$^2$ = 0.980 and 0.996 respectively) across an appropriate range of concentrations for the peak areas in the samples. The gradient obtained for the 2-MT-OS standard was ~4 times higher than

that of the 2-MG-OS, as shown in Figure S2. To investigate the potential for matrix effects from the large amounts of inorganic sulfate, nitrate and other particulate components that co-elute due to the poor retention of OS in reverse phase UHPLC, standard addition calibrations were used. Five-point standard addition calibrations were run on 6 different filter extracts, covering both day and night-time, samples, during periods of both high and low concentrations of iSOA species. This therefore gives a representative

sample of filters for the entire sampling period. 50 μL of filter sample extract and 50 μL of the calibrant solution were combined, giving a dilution factor of 2. The five-point calibration range of standard added to each sample was between 0-3 ppm for 2-MGOS and 0-1 ppm for 2-MT-OS. Two examples of the standard addition calibrations are shown in SI Figures S3 (2-MG-OS) and S4 (2-MT-OS), with good linear fits observed (R$^2$ = 0.997 and 0.997 respectively). A strong matrix effect was observed for the 2-

MT-OS, with the concentration measured by standard addition calibration 8.6 to 10 times higher than when using the external calibration carried out on the same day. In contrast, the 2-MG-OS showed a much lower matrix effect, with the concentrations only 1.1-1.5 times higher when using the standard addition calibration. A further comparison using camphorsulfonate, which has a longer retention time (3.74 min) and so does not experience high inorganic ion concentrations in the source, showed no matrix

effects when using standard addition. Tables S1 and S2 shows a comparison of the concentrations calculated from the standard additions and the two external calibrations. Table S1 shows the concentration of 2-MT-OS in three filter sample extracts (144, 204, 208) calculated via standard addition of 2-MT-OS to the filter sample extract and via external calibrations using both 2-MT-OS and 2-MG-OS. The ratio of the standard addition to the external calibrations then gives an estimate of the under or

overestimate the external calibrations make to calculating the concentration of 2-MT-OS in the samples. Both the external calibrations would lead to an underestimation of concentration of 2-MT-OS in the filter

samples. 2-MG-OS provided a closer quantification of 2-MT-OS in the samples, with an average factor of 2.3 underestimation, while the 2-MT-OS external calibration gives a sample concentration a factor of 10 lower than the standard addition determined concentration.

It is not realistic to carry out standard addition calibrations for all samples and all SOA tracers. When the 2-MG-OS external calibration was used to predict the 2-MT-OS concentrations during the standard addition experiments, the concentrations were within a factor of 1.5-2.5. Therefore, the 2-MG-OS external calibration was used as a proxy for all isoprene SOA tracers, with scaling factors applied to account for matrix effects (1.33 for 2-MG-OS, 2.33 for 2-MT-OS, and an average of 1.83 used for all

other OSs). Therefore, we estimate an uncertainty on our measured concentrations of 60%, this uncertainty was calculated to account for the difference in the measured correction factors used when correcting for the matrix effects. The uncertainty was calculated as $2\sigma$ of the 6 values used to calculate the average correction factor of 1.83. The matrix effects identified in this study are likely due to the extracted samples being a complex mixture of different compounds, including a high proportion of

inorganic ions that are extracted into water. This is likely to change the surface tension of the droplet produced in the ionisation source and the ion distribution. Further work is needed to fully understand the reasons. Without these additional standard addition calibrations, the iSOA concentrations would have been largely underestimated. The dinitrate and trinitrate NOS species eluted after the sulfate peak ($R_t$ >1.6 min). In the absence of authentic standards for these species, camphorsulfonic acid was used as a

proxy for calibration. This work highlights an additional difficulty of calibration when using ESI-MS to study OSs and indicates that future studies using reversed phase LC (RPLC) should consider the impacts of matrix effects.

### 3.5 Organosulfates

**3.5.1 2-methyltetrol OS (2-MT-OS)**

The 2-MT-OS ($C_5H_{12}SO_7$) formed from the uptake of IEPOX into the particle phase is often used as a marker of low-NO isoprene photochemistry (Wennberg et al., 2018). The time series of 2-MT-OS is shown in Figure 6a. The particle concentration ranged from 0.7 ng m$^{-3}$ to a maximum of 111 ng m$^{-3}$, with a mean concentration of 11.8 ng m$^{-3}$. The mean concentrations of 2-MT-OS and 2-MG-OS are compared

to observations in previous studies in Table 3. The mean concentration observed in Beijing was much lower than those observed in the Amazon (Riva et al., 2019) and the SE US (Budisolistiorini et al., 2015; Hettiyadura et al., 2019) but are higher than summer time observations at polluted regional sites in China (Wang et al., 2018; He et al., 2018). The lower amounts of IEPOX-derived SOA results in an average AMS $f_{C5H6O}$ in Beijing during the APHH project of only 0.2 %, similar to observations in other urban

studies (Hu et al., 2015).

Hourly samples were collected on selected high pollution days and used to obtain information on the diurnal evolution of the iSOA tracers. The findings on these days are consistent with the three-hourly data. The particulate 2-MT-OS measured by UHPLC-MS, on the 11$^{th}$ - 12$^{th}$ June 2017, had a strong

diurnal profile (Figure 3b), peaking in the late afternoon, between 15:30 and 18:30, with a minimum over-night. This is consistent with the average diurnal profile of the gas phase precursors

IEPOX+ISOPOOH ($C_5H_{12}O_3$) measured using the $I^-$-CIMS (SI Figure S5). High levels of ozone were observed in the afternoon (up to 180 ppb), leading to relatively low levels of NO observed for a highly polluted environment, in some cases below 500 ppt. Thus, although the mixing ratio of $NO_x$ was high, on most afternoons less than 2 % was in the form of NO. High levels of peroxy radicals were observed, with mean afternoon concentrations of $HO_2$ and $RO_2$ of around 3 x $10^8$ molecule $cm^{-3}$ and 1.5 x $10^9$ molecule $cm^{-3}$, respectively. Zero-dimensional box modelling indicates on some days up to 35 % of the isoprene-derived $RO_2$ radicals can react with $HO_2$ in the afternoon (Newland et al., 2019). Thus, the diurnal profile seen in Figure 3b, measured in samples during the measurement period suggests that IEPOX was formed at this urban location by the reaction of OH with local isoprene emissions, with a fraction of the $RO_2$ radicals formed reacting with $HO_2$ rather than NO, and subsequent uptake to aerosol forming 2-MT-OS. OH + isoprene hydroxynitrate also has a small yield of IEPOX (Jacobs et al., 2014). The average diurnal profile of isoprene hydroxynitrates ($C_5H_9NO_4$) in the gas phase measured using the $I^-$-CIMS peaks at around 11:00-12:00 followed by a reduction during the afternoon into the evening/night (SI Figure S6). This is likely to be a result of the relatively low levels of NO during the afternoon, which will reduce isoprene nitrate formation from $RO_2$ + NO reactions, thus isoprene hydroxynitrates are unlikely to be a significant source of 2-MT-OS in Beijing.

The 2-MT-OS showed a moderate correlation with particulate sulfate ($R^2$=0.44), and a weak anti-correlation with photochemical age, estimated using the ratio of $NO_x/NO_y$ ($R^2$=0.23) as shown in Figure 5. All correlations between species are shown in Figure 5. By taking the product of the concentration of ozone, as a proxy of photochemistry, with the amount of particulate sulfate measured using AMS, $[O_3][pSO_4]$, a much stronger correlation with 2-MT-OS was observed ($R^2$=0.61) as shown in Figure 7. This observation highlights the role of both local photochemistry and particulate sulfate mass in the formation of 2-MT-OS (Figure 7). The correlation of $[O_3][pSO_4]$ with 2-MT-OS is likely to be weaker at longer photochemical ages when the ozone concentration is not directly related to the photochemical formation of the OS. Again, this highlights the strong role of local photochemistry in the production of low-NO iSOA (2-MT-OS) in Beijing. Elevated levels of 2-MT-OS were observed at the start and end of the measurement period which were influenced by strong south-westerly winds. There were also elevated isoprene concentrations (up to 2.9 ppb) and high particulate $SO_4^{2-}$ levels. Therefore, these spikes in 2-MT-OS could be a result of either higher 2-MT-OS in regional aerosol transported to the site or a high isoprene emission source to the south west of the site (i.e. producing IEPOX locally) that then reacts with increased regional sulfate pollution. The I-CIMS data shows that the IEPOX/ISOPOOH (Figure S5 and Newland et al., 2020) signal increases during the afternoon as the NO levels drop to below 1 ppb. The low NO levels mean that up to 30 % of the isoprene peroxy radical from OH oxidation can react with $HO_2$ rather than NO at this site, meaning IEPOX can be formed locally (Newland et al., 2020). There is also likely to be a regional source of IEPOX and 2-MT-OS, suggesting both local and regional anthropogenic influences.

Analysis of the 2-MT-OS isomer distribution using HILIC/ESI-HR-QTOFMS, on a subset of 15 samples,

indicates that β-IEPOX is the dominant ambient IEPOX isomer, in line with other recent observations (Cui et al, 2018; Krechmer et al., 2016) see SI Figure S7). The MT-OS derived exclusively from δ-IEPOX-OS isomers could not be observed in any of the samples. The 4 IEPOX-OS isomers in SI Figure S7 showed similar temporal trends although small changes in the relative proportions were observed. The sum of peak areas from the 2-MT-OS isomers measured by HILIC and the quantified 2-MT-OS (sum of isomers) measured via UHPLC/ESI-HR-MS were compared and showed a high degree of correlation ($R^2 = 0.84$), even though the two methods used different solvents. The agreement indicates that the UHPLC/ESI-HR-MS method captures the sum of the isomers and there is no evidence of ion source induced artefacts.

### 3.5.2 2-methyl glyceric acid OS (2-MG-OS)

The most common targeted SOA tracer for high-NO isoprene chemistry is 2-methylglyceric acid (2-MG) and its derivatives. As such, this tracer is the result of a direct biogenic-anthropogenic interaction. Two observed SOA tracers related to this chemistry are the OS derivatives of 2-methylglyceric acid (2-MG-OS) and the unresolved $C_8$ dimers of 2-MG-OS ($C_8H_{14}SO_{10}$) that have been identified previously in chamber-derived iSOA (Surratt et al., 2006; Surratt et al., 2010). 2-MG-OS had an average concentration during the campaign of 21.5 ng m$^{-3}$, ranging from 0.3 to 180.5 ng m$^{-3}$, with the time series shown in Figure 6b. These values are within the range of 2-MG-OS measured in other urban locations (Nguyen et al., 2014; Rattanavaraha et al., 2016; Hettiyadura et al., 2019). However, these concentrations are considerably higher than previously observed at two Chinese regional background sites (Wang et al., 2018; He et al., 2018). At these locations, the ratio of the low-NO to high-NO isoprene OS tracer average concentrations was close to 1.5 (2-MT-OS:2-MG-OS; Beijing = 1.47, Wanqingsha = 1.57). However, in central Beijing, this ratio was considerably lower (2-MT-OS:2-MG-OS = 0.55), reflecting the higher proportion of RO$_2$ radicals reacting with NO at this location compared to the regional measurements. The ratio of 2-MT-OS:2-MG-OS observed in Beijing is compared to previous studies in Table 23 and is considerably lower than measurements taken in a range of isoprene dominated environments (South East US, 2-MT-OS:2-MG-OS = 17, Budisolistiorini et al., 2015.; Amazon, 2-MT-OS:2-MG-OS = 13-118, Glasius et al., (2018).; Atlanta, 2-MT-OS:2-MG-OS = 33, Hettiyadura et al., (2019)) reflecting the strong impact of urban NO emission on iSOA formation. Future work will investigate how to use these ratios to quantify the effect of NO emission on iSOA formation in different regions.

The mean concentration of the 2-MG-OS dimer ($C_8H_{14}SO_{10}$) was 0.57 ng m$^{-3}$. A strong linear relationship was observed between the 2-MG-OS monomer and dimer concentrations ($R^2=0.83$) with a dimer:monomer ratio of 0.02. Formation of oligomers from reactions of 2-MG and HMML has been shown to be reduced in chamber experiments under humid conditions (Schwantes et al., 2019; Nestorowicz et al., 2018). The average RH during the afternoon of the campaign was ~40 %, which may account for the relatively low formation of the dimer OS compared to the monomer (see SI Figure 89).

The diurnal profile of the 2-MG-OS as shown in Figure 8 was similar to the 2-MT-OS peaking during the early afternoon samples but with an enhanced signal at night. There was also a strong correlation between these two species ($R^2 = 0.92$) during the campaign. The 2-MG-OS showed a stronger correlation with particulate sulfate ($R^2=0.52$) than 2-MT-OS ($R^2=0.44$), and there was also a weak anti-correlation with photochemical age ($R^2=0.28$). A strong correlation was also observed for 2-MG-OS with $[O_3][pSO4]$ ($R^2=0.69$), as shown in Figure 7, highlighting that formation is dependent on both photochemistry and sulfate aerosol availability.

### 3.5.3 Other isoprene-related OSs and NOSs

24 additional OSs species, with molecular formulae consistent with iSOA tracers seen in chamber experiments, were also observed in Beijing as shown in Table 2. For $C_5$ compounds, the most abundant species were $C_5H_{10}SO_6$ and $C_5H_{10}SO_5$, with mean concentrations of 28.7 ng m$^{-3}$ and 26.5 ng m$^{-3}$, respectively. The identity of the OS at $m/z$ 182 ($C_5H_{10}SO_5$) is currently unknown and the product ion MS provides little additional information other than sulfate-related fragment ions at $m/z$ 97 and $m/z$ 80. The OS at m/z 198 ($C_5H_{10}SO_6$) was identified as an IEPOX-related OS in chamber experiments by Nestorowicz *et al.* (2018), but at relatively low concentrations compared to the 2-MT-OS (1-4 %). This is very different to the observed ratio is Beijing, where the $C_5H_{10}SO_6$ average concentration was more than double that of 2-MT-OS, as shown in Figure 6c. This compound showed a strong correlation with 2-MT-OS ($R^2 = 0.77$) but it is currently unclear why this compound is the most abundant $C_5$ species. The molecular weight of this species is 18 Da (-H$_2$O) lower than 2-MT-OS, which may indicate it is a dehydration product enhanced under acidic aerosol conditions. In addition, this species may also be enhanced if it is formed from additional VOC precursors.

Potential low-NO iSOA tracers, seen in chamber experiments, correlated strongly with the 2-MT-OS including unresolved isomers of cyclic hemiacetals [$C_5H_{10}SO_7$ ($R^2=0.92$)], and lactones [$C_5H_8SO_7$ ($R^2=0.83$)] (Spolnik et al., 2018). These compounds were similar in concentration to the 2-MT-OS, with the lactones at MW 212 having a mean concentration of 14 ng m$^{-3}$ and the cyclic hemiacetals at MW 214 a mean of 10.6 ng m$^{-3}$. These compounds were also observed to be the dominant type of isoprene-derived OSs in Atlanta, Georgia, although they had concentrations a factor of ~15 times lower than the observed 2-MT-OS. (Hettiyadura et al., 2019)

Additional small OS compounds, previously identified during high-NO chamber experiments, were also observed in Beijing, including in order of decreasing concentration, glycolic acid sulfate ($C_2H_4SO_6$, mean = 38.4 ng m$^{-3}$), hydroxyacetone sulfate ($C_3H_6SO_5$, mean = 20.5 ng m$^{-3}$) and lactic acid sulfate ($C_3H_6SO_6$, mean = 14.5 ng m$^{-3}$) (Surratt et al., 2007; Surratt et al., 2008). These concentrations are in line with measurements made in other urban locations (Rattanavaraha et al., 2016; Huang et al., 2018; Hettiyadura et al., 2018). While all three $C_2$-$C_3$-OS compounds had strong correlations with the other iSOA OS tracers ($R^2 = 0.6$-$0.94$), the relative strength of isoprene versus other VOC precursors, such as aromatics, cannot be determined. As such, they cannot be definitively assigned as iSOA tracers, and are therefore included

in the potential iSOA portion of Figure 9. The sum of the $C_2$ and $C_3$ OSs had an average concentration of 73 ng m$^{-3}$, with a range of 2.0-831 ng m$^{-3}$.

In addition, 9 NOS species related to isoprene were identified as shown in Table 2 (Ng et al., 2008; Rollins et al., 2009). Some of the NOS observed peaked in the daytime and some were enhanced at night. In total they had a mean concentration of 24 ng m$^{-3}$ during the campaign. NOS species are formed via the heterogenous uptake of isoprene nitrates (IN) into the particle phase. Nitrate radicals play a key role in the formation of IN, with nitrate radicals forming from reaction of $NO_2$ with $O_3$, both key anthropogenic pollutants. Therefore, emissions of NOx and formation of particulate sulfate will enhance the production of isoprene NOS species.

### 3.6 Contribution of Isoprene SOA in Beijing

In order to estimate the total amount of isoprene-derived OSs and NOSs, labelled here as iSOA, 13 species were chosen that could be confidently identified as being predominately from isoprene (2-MT-OS, 2-MG-OS, $C_5H_{10}SO_7$, $C_5H_8SO_7$, $C_5H_{11}NSO_9$, $C_5H_9NSO_{10}$, $C_5H_9N_2SO_{11}$, $C_5H_8N_3SO_{13}$). Although there were a number of other compounds with formula similar to iSOA tracers, their trends compared to previous studies and potential for alternative sources made a confident assignment of VOC precursor difficult. Therefore, the estimated contribution of iSOA to the observed total particulate mass determined here should be taken as a lower limit. Figure 9 shows the time series of the iSOA observed in Beijing. The average concentration was 82.5 ng m$^{-3}$ during the campaign, ranging from 718 ng m$^{-3}$ on the 19/05/2017 (11:38 – 14:30) to 1.9 ng m$^{-3}$ on the 02/06/2017 (14:36-17:28). The contribution of iSOA to the OOA factors measured by the AMS was obtained by assuming all OSs and NOSs species fragment in the ion source to lose the sulfate and nitrate groups. Across the whole measurement period, the iSOA tracers represented only a small fraction of the total OOA measured by AMS (0.62% of $\sum$[OOA1-3]). However, towards the end of the measurement period, this increased up to a maximum of 3 % on the 17/06/2017 (13:32-14:23).

Additional iSOA tracers containing only CHO (Table 3), including 2-methyltetrols, 2-methyglyceric acid and $C_5$-alkene triols, were measured in separate 24-hour filter samples, with the commonly used derivatization GC-MS method (Claeys et al., 2004; Wang et al., 2005.). The average ratio of the 2-methyltetrols to its corresponding OS (2-MT:2-MT-OS) was 1.4, indicating extensive heterogeneous conversion of isoprene oxidation products within the particles. The observed ratio is slightly larger than those measured in the SE US (~0.37-0.96 as shown in Table 3) but much lower than that measured in the Pearl River Delta region (~40,) where the 2-methyltetrols dominated. In contrast, the average ratio of the high-NO iSOA tracer, 2-MG and its corresponding organosulfate (2-MG:2-MG-OS) observed in Beijing was 0.33, indicating more extensive transformation to products from heterogeneous reactions. This ratio may also reflect the more volatile nature of 2-MG compared to 2-MT. Overall, the combined concentrations of these isoprene CHO compounds were generally low (mean 25 ng m$^{-3}$, max 69 ng m$^{-3}$) in comparison to the heterogeneous iSOA compounds (i.e., isoprene-derived OSs and NOSs) targeted in this work. In addition, the concentrations of these CHO species may be overestimated based on recent

studies demonstrating that thermal decomposition leads to these products being detected by GC-MS and FIGAERO-CIMS methods (D'Ambro et al., 2019), and so the conversion to products from heterogenous reactions (i.e., OSs and NOSs) may in fact be larger (2MT:2MT-OS = 0.5-0.91 using the overestimates of 160-288 % observed in Cui et al., (2018)).

The study presented here shows for the first time that OS species derived from isoprene oxidation can make a significant contribution to oxidised organic aerosol in Beijing in summer. There is significant anthropogenic control, from both $NO_x$ and sulfate aerosols, on the products and concentrations of iSOA in Beijing. The majority of the OS species showed a strong correlation towards the product of $[O_3][pSO_4]$, highlighting the role of both photochemistry and the availability of particulate sulfate for heterogeneous

reactions. When the observed concentrations of all the OS and NOS species measured in this study, including the additional 19 compounds not confidently assigned to iSOA, are combined they contribute on average 2.2 % to the total OOA ($\sum$[OOA1-3]), increasing to a maximum of 10.5 %, indicating extensive heterogeneous conversion of VOC oxidation products in Beijing in summer.

**Author contributions**

DB analysed the aerosol samples and quantified iSOA tracers. WD and KP developed the UHPLC-MS method. JRH, RD and MS provided the VOC measurements. FS and JL collected the NO, $NO_2$ and $O_3$ data. TB, AM, SW, AB, CJP, HC collected and analysed the CIMS data. LW, DH and ES provided the OH and $HO_2$ data and BO provided the $NO_3$ measurements. TC, JDS and WD carried out the offline

HILIC analysis. DL, ZS and RH provided the GC-MS iSOA data. YS and WX carried out the AMS measurements and PMF analysis. ACL and RH lead the APHH projects. DB, ARR and JFH wrote the manuscript with input and discussion with all co-authors.

**Competing interests**

The authors declare that they have no conflict of interest.

**Data availability**

Concentration data is available upon request from JRH. Filter collection times are shown in Table S3

**Acknowledgements**

This project was funded by the Natural Environment Research Council, the Newton Fund and Medical Research Council in the UK, and the National Natural Science Foundation of China (NE/N007190/1, NE/N006917/1). We acknowledge the support from Pingqing Fu, Zifa Wang and Jie Li from IAP for hosting the APHH-Beijing campaign at IAP. We thank Tuan Vu and Bill Bloss from the University of

Birmingham, Siyao Yue, Liangfang Wei, Hong Ren, Qiaorong Xie, Wanyu Zhao, Linjie Li, Ping Li, Shengjie Hou, Qingqing Wang from IAP, Kebin He and Xiaoting Cheng from Tsinghua University, and James Allan from the University of Manchester for providing logistic and scientific support for the field campaigns. Daniel Bryant, Freya Squires, William Dixon and Eloise Slater acknowledge NERC SPHERES PhD studentships. Marvin Shaw acknowledges SYFT Technologies for his fellowship grants

and scientific support. The Orbitrap-MS was funded by a Natural Environment Research Council strategic capital grant, CC090. Jason Surratt and Tianqu Cui acknowledge support from the United States National Science Foundation (NSF) under Atmospheric and Geospace (AGS) Grant 1703535 as well as thank Avram Gold and Zhenfa Zhang from the University of North Carolina for providing the 2-MT-OS and 2-MG-OS standards.

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

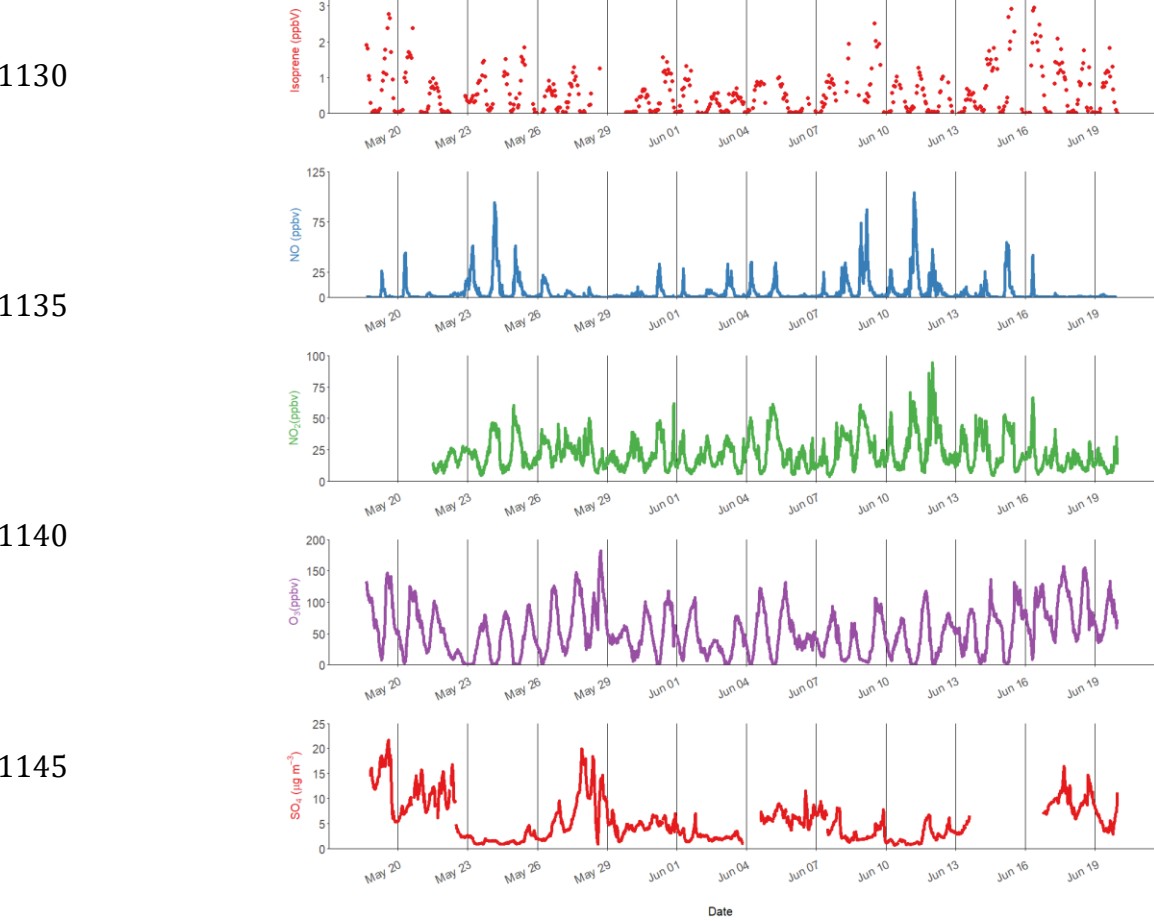





**Figure 1.** Time series of isoprene, nitric oxide (NO), nitrogen dioxide (NO$_2$), ozone (O$_3$) and particulate sulfate (SO$_4$). The black lines are at midnight every 72 hours.

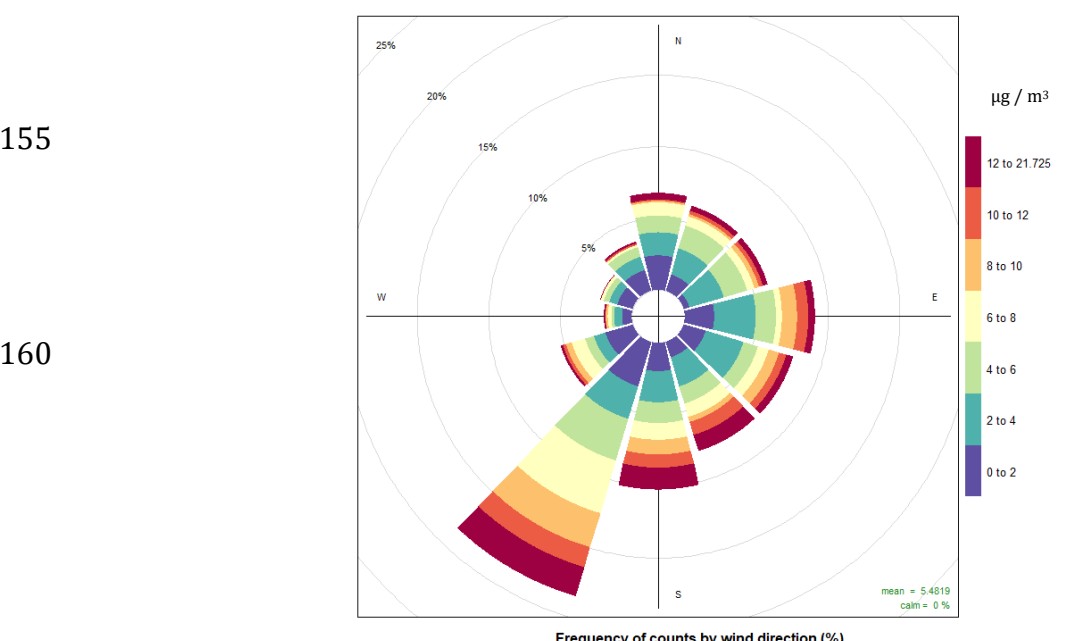



**Figure 2.** PollutionRose plot (Openair) of particulate sulfate measured by AMS, for the sampling period. Highlighting under what wind conditions the highest concentrations of sulfate occur.

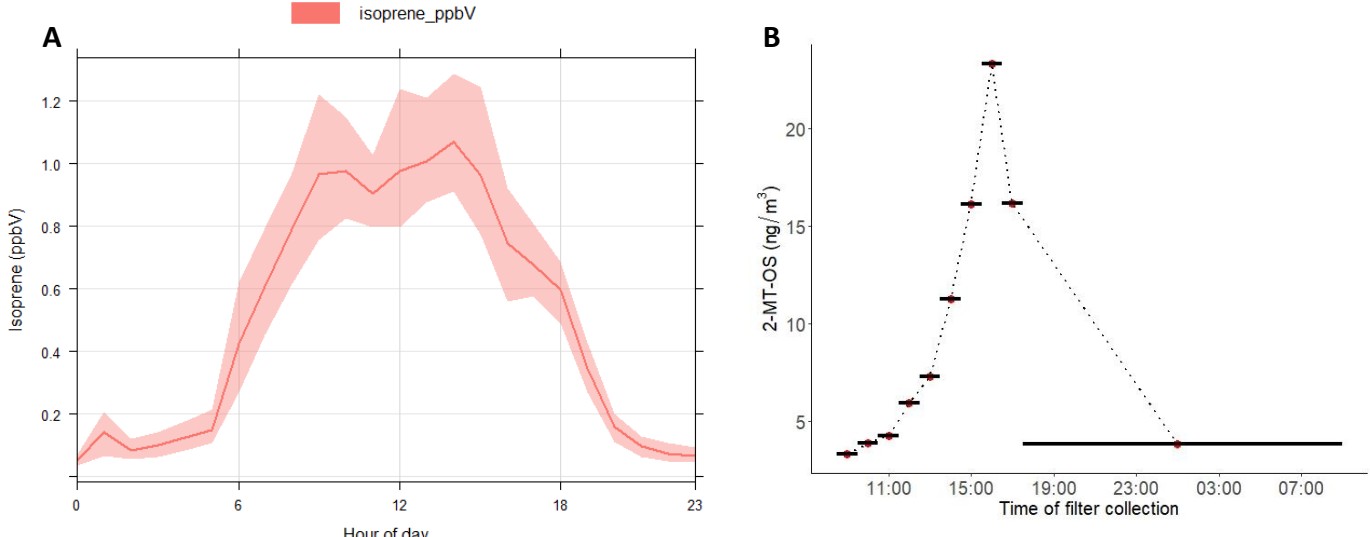

**Figure 3.** (A) Average diurnal profile of isoprene mixing ratio measured using DC-GC-FID. (B) Diurnal profile of 2-methyltetrol sulfate (2-MT-OS) in particulate matter ($PM_{2.5}$) collected on filters hourly over the 11[th] to 12[th] June 2017. Black lines indicate length of filter sampling period.













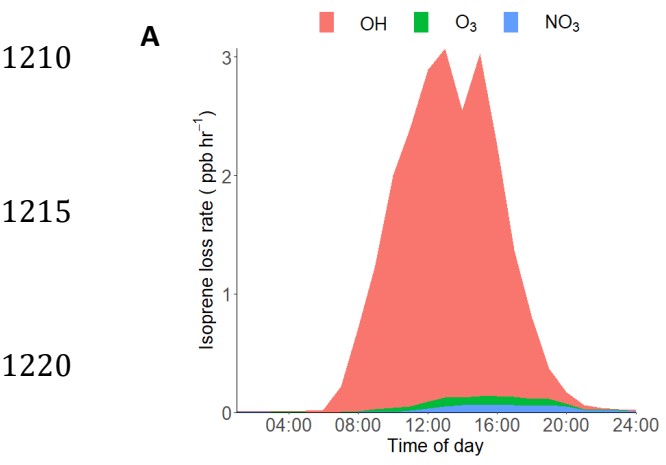
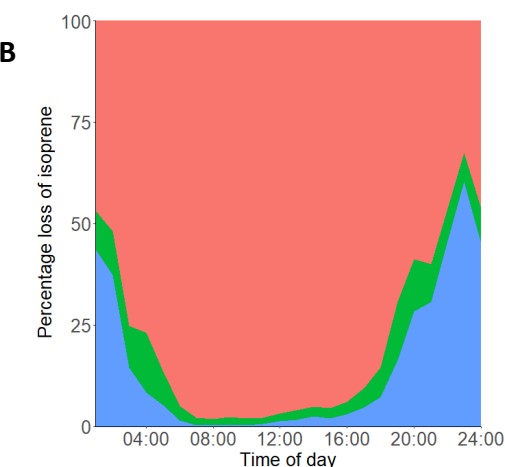


**Figure 4.** (A) Diurnal loss rate of isoprene calculated using measured average diurnal profiles of isoprene, OH, NO$_3$ and ozone. (B) Average diurnal of the percentage loss of isoprene from reactions with OH, O$_3$ and NO$_3$ radicals. The IUPAC rate constants used for the calculations are as follows, NO$_3$: $7 \times 10^{-13}$, O$_3$:$1.27 \times 10^{17}$, OH: $1 \times 10^{10}$ cm$^3$ molecule$^{-1}$ s$^{-1}$ (Atkinson et al., 2006).



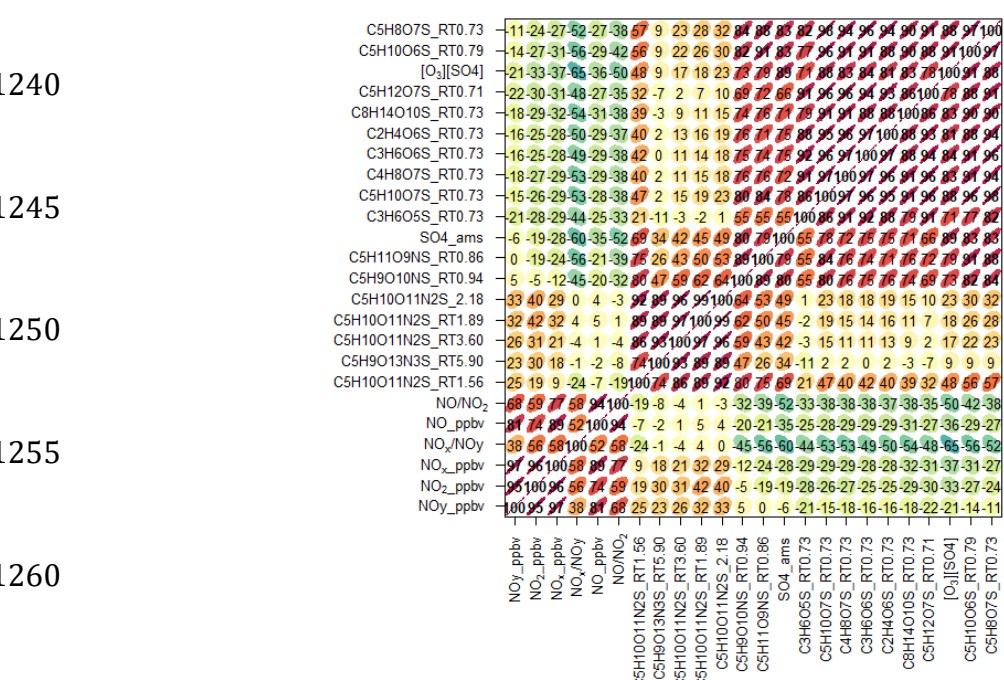







**Figure 5.** Correlation plot (R, Openair, CorPlot) highlighting the correlations between known iSOA tracers and anthropogenic pollutants. The number represents the R correlation between the two species. With redder more elongated circles highlighting a higher correlation.






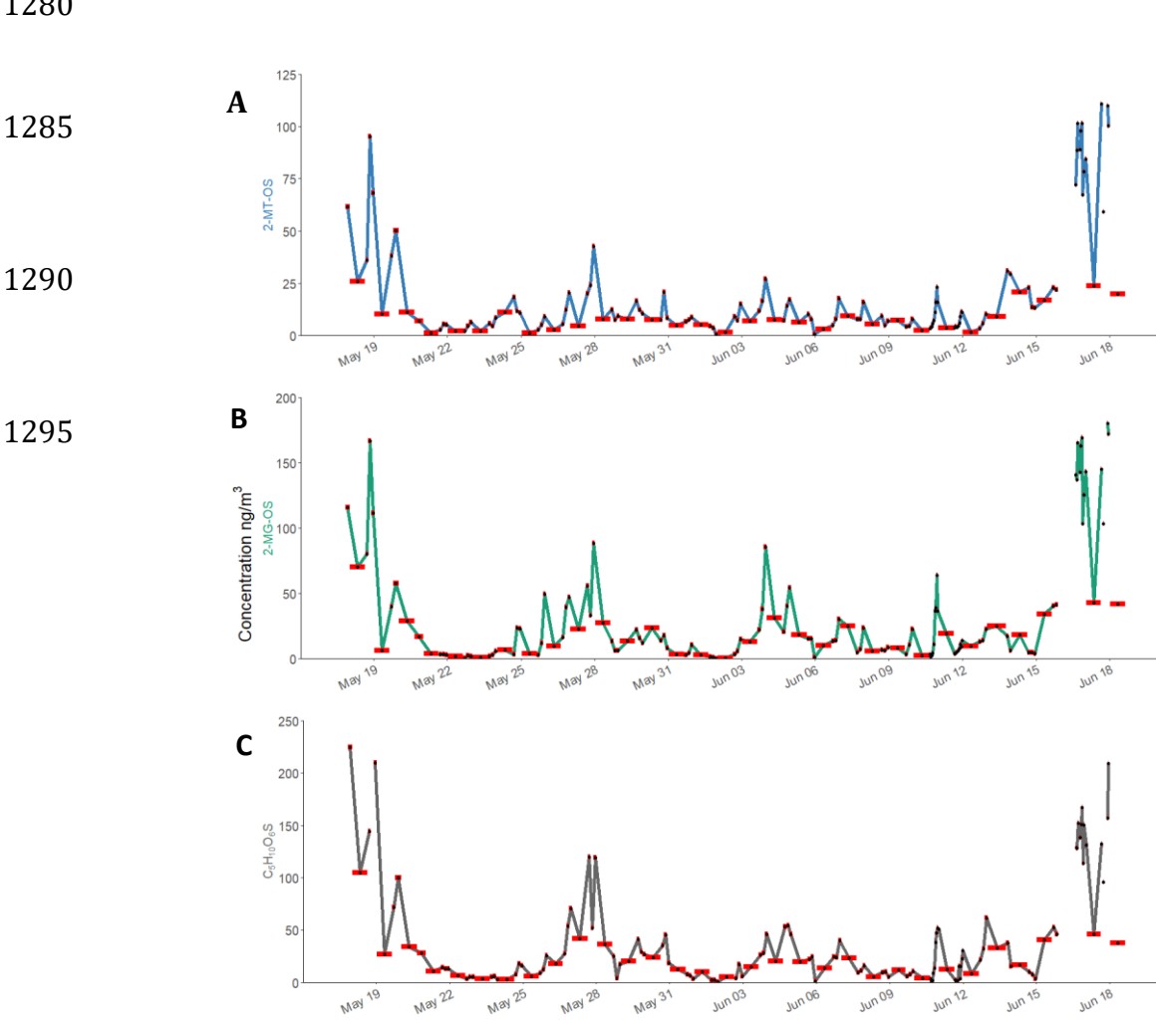


**Figure 6.** Time series of observed concentrations of iSOA tracers in Beijing during APHH. (A) 2-MT-OS ($C_5H_{12}O_7S$) (B) 2-MG-OS ($C_4H_8O_7S$) (C) $C_5H_{10}SO_6$. The red bars indicate the length of the sampling time. The individual sample times will be given in the accompanying dataset. *(doi to be given on acceptance)*


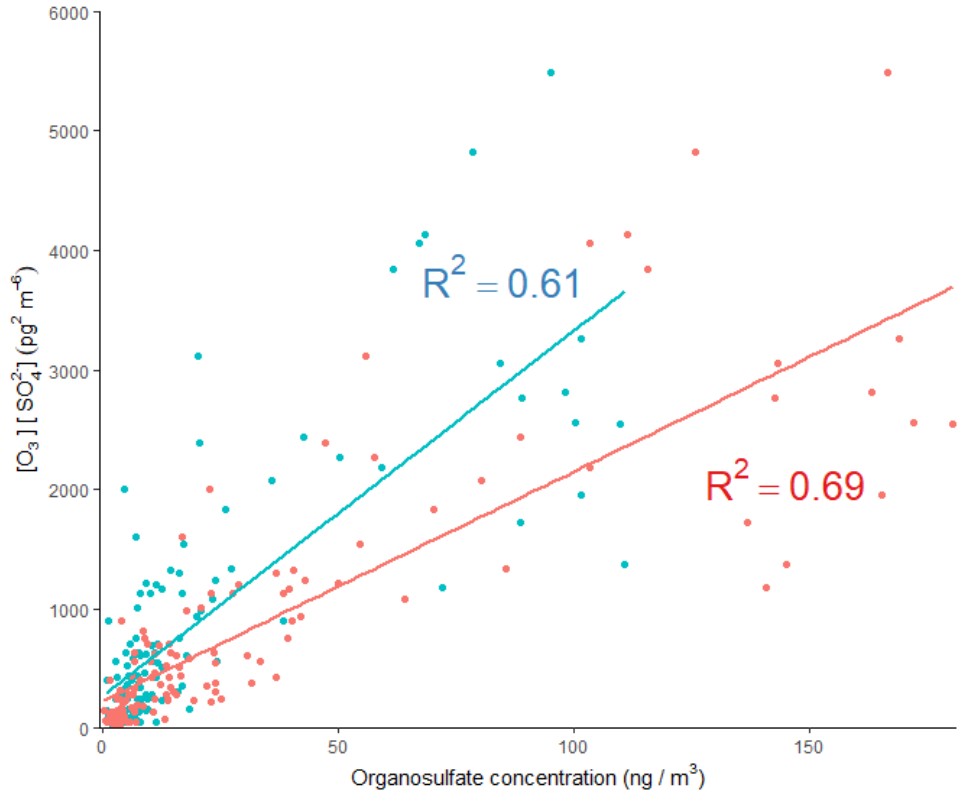


**Figure 7**. Plot of 2-MT-OS ($C_5H_{12}O_7S$, blue) and 2-MG-OS ($C_4H_8O_7S$, red) concentrations versus [Ozone][$SO_4$]. The high time resolution data ($O_3$ and AMS $SO_4^{2-}$) has been averaged to the filter sampling time. The line was calculated using the stat_smooth function in the R package ggplot2, using the method "lm".


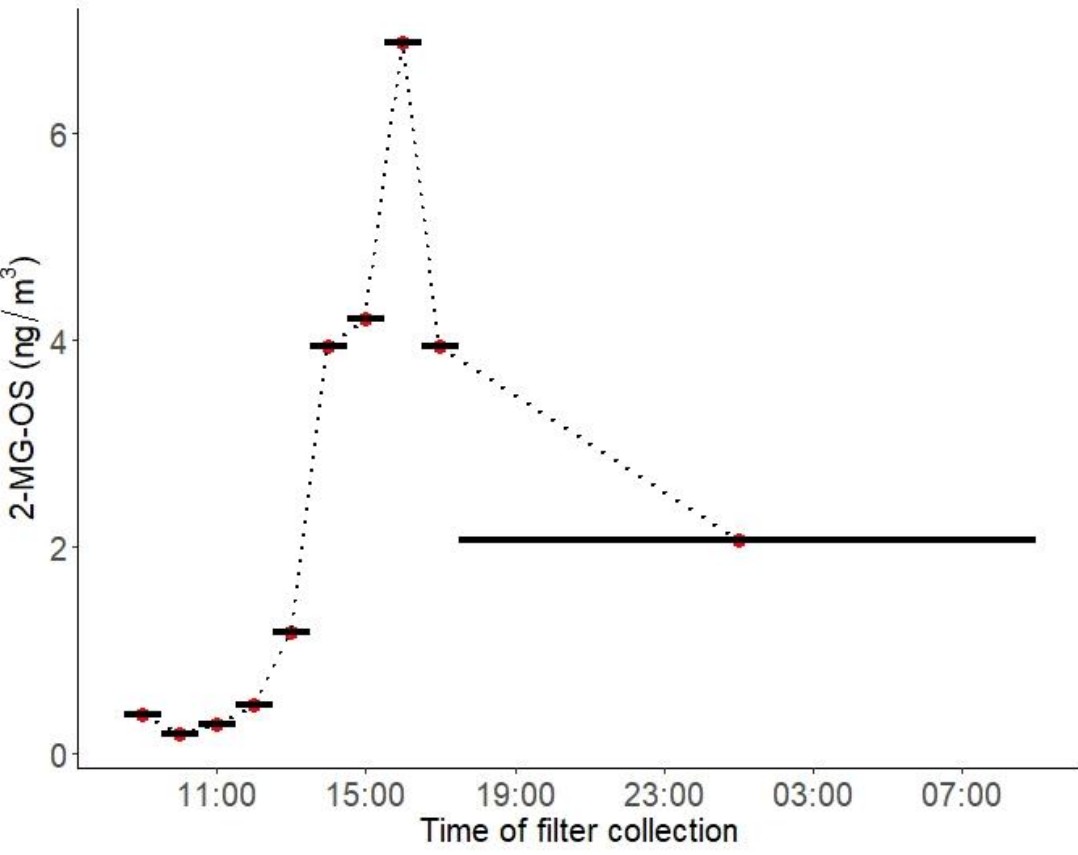

**Figure 8.** Diurnal profile of 2-methylglyceric acid sulphate (2-MG-OS) in particulate matter collected on filters hourly over the 11[th] to 12[th] June 2017. Black lines indicate length of sampling.

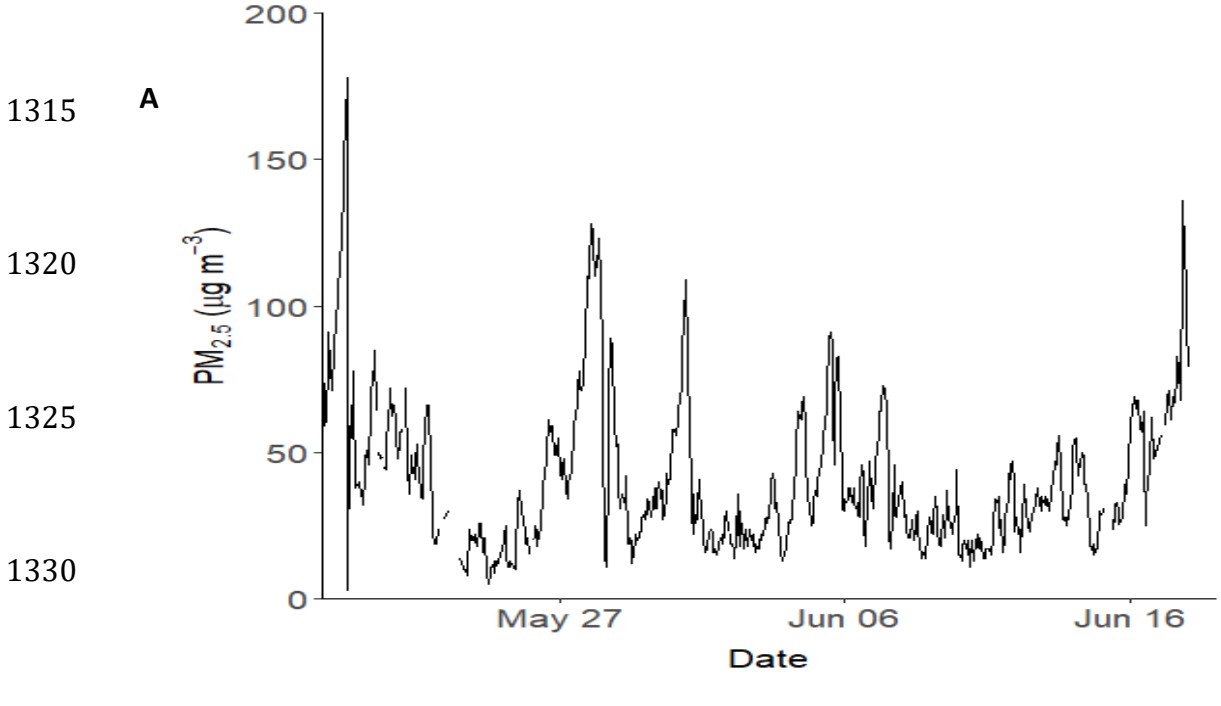

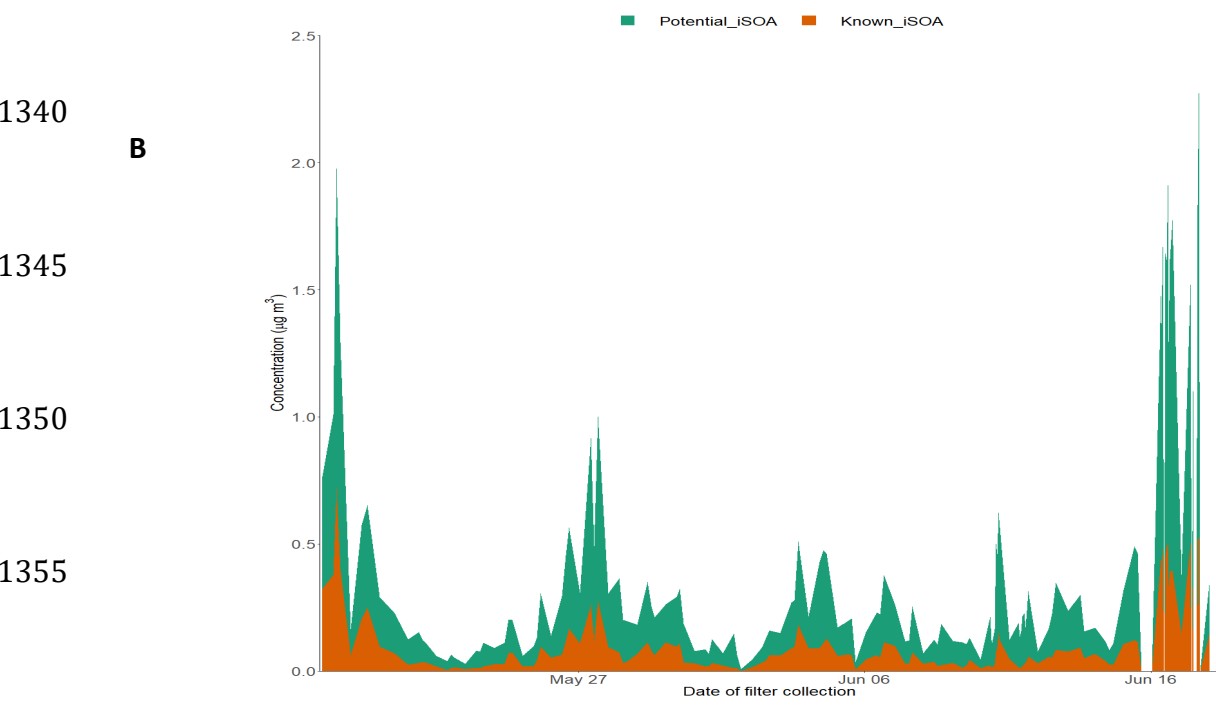

**Figure 9.** A) Time series of PM$_{2.5}$ over the sampling period. B) Time series of the total known isoprene SOA signal (2-MT-OS, 2-MG-OS, C$_5$H$_{10}$SO$_7$ (MW 214) C$_5$H$_8$SO$_7$ (MW 212) C$_5$H$_{11}$NSO$_9$ (MW 261), C$_5$H$_9$NSO$_{10}$ (MW 275) C$_5$H$_{10}$O$_{11}$N$_2$S (MW 306), C$_5$H$_9$O$_{13}$N$_3$S (MW 351) and the total signal from the other iSOA tracers quantified in this study.


**Table 1.** Supplementary anthropogenic pollutants measured during the sampling period analysed in this study.


| Pollutant | Mean ± SD | Max | Min |
|---|---|---|---|
| $O_3$ (ppbv) | 54.0 ± 37.5 | 181.8 | 2.0 |
| NO (ppbv) | 5.1 ± 11.3 | 104.1 | 0.1 |
| $NO_2$ (ppbv) | 22.3 ± 13.0 | 94.5 | 3.7 |
| $SO_4$ ($\mu g/m^{-3}$) | 5.5 ± 4.1 | 21.7 | 0.7 |











**Table 2.** Molecular formulas, negative ion masses, retention times (RT), time weighted means (ng m$^{-3}$) for the entire sampling period and original reference to where the tracer was found of each proposed iSOA tracer. BD = Below detection. The estimated uncertainties were discussed in section 3.4 as 60 %, accounting for the use of the matrix correction factors.


| Isoprene Tracer | [M-H]$^{-1}$ | RT (min) | Time weighted mean (ng m$^{-3}$) | Maximum (ng m$^{-3}$) | Minimum (ng m$^{-3}$) | Reference |
|---|---|---|---|---|---|---|
| $C_2H_4O_6S$ | 154.9656 | 0.73 | 38.4 | 366.1 | BD | Surratt et al., 2008 |
| $C_5H_{10}O_6S$ | 197.0125 | 0.79 | 28.7 | 336.2 | 0.25 | Surratt et al., 2007 |
| $C_5H_{10}O_5S$ | 181.0176 | 0.93 | 26.5 | 448.5 | 2.91 | Nguyen et al., 2010 |
| $C_4H_8O_6S$ | 182.9969 | 0.73 | 21.7 | 229.1 | 0.50 | Riva et al., 2016 |
| $C_4H_8O_7S$ | 198.9918 | 0.73 | 21.5 | 180.5 | 0.32 | Surratt et al., 2007 |
| $C_3H_6O_5S$ | 152.9863 | 0.73 | 20.5 | 327.9 | 0.98 | Surratt et al., 2008 |
| $C_3H_6O_6S$ | 168.9812 | 0.73 | 14.5 | 137.7 | 0.25 | Surratt et al., 2008 |
| $C_5H_8O_7S$ | 210.9918 | 0.73 | 14.0 | 136.4 | 0.27 | Surratt et al., 2008 |
| $C_5H_{11}O_9NS$ | 260.0082 | 0.86 | 12.6 | 154.1 | 0.10 | Surratt et al., 2008 |
| $C_5H_{12}O_7S$ | 215.0231 | 0.71 | 11.8 | 110.9 | 0.77 | Surratt et al., 2008 |
| $C_5H_{10}O_7S$ | 213.0075 | 0.73 | 10.6 | 104.7 | 0.38 | Surratt et al., 2008 |
| $C_5H_9O_{10}NS$ | 273.9874 | 0.94 | 9.17 | 53.8 | BD | Nestorowicz et al., 2018 |
| $C_4H_8O_5S$ | 167.0019 | 0.73 | 9.10 | 114.5 | 0.68 | Surratt et al., 2007 |
| $C_5H_8O_5S$ | 179.0020 | 0.85 | 6.59 | 144.2 | 0.43 | Riva et al., 2016 |
| $C_5H_{10}O_5S$ | 181.0176 | 1.24 | 4.90 | 36.3 | 1.21 | Riva et al., 2016 |
| $C_5H_{10}O_8S$ | 229.0024 | 0.75 | 4.59 | 40.9 | BD | Nestorowicz et al., 2018 |
| $C_5H_8O_9S$ | 242.9816 | 0.64 | 1.55 | 13.9 | BD | Nestorowicz et al., 2018 |
| $C_5H_{10}O_{11}N_2S$ | 304.9783 | 2.18 | 1.04 | 8.62 | BD | Surratt et al., 2008 |
| $C_{10}H_{20}O_8S$ | 299.0806 | 1.65 | 1.01 | 8.38 | BD | Riva et al., 2016 |
| $C_5H_{10}O_{11}N_2S$ | 304.9783 | 1.89 | 0.83 | 7.69 | BD | Surratt et al., 2008 |
| $C_8H_{14}O_{10}S$ | 301.0235 | 0.73 | 0.57 | 4.16 | BD | Surratt et al., 2007 |
| $C_5H_{10}O_{11}N_2S$ | 304.9783 | 1.56 | 0.42 | 2.90 | BD | Surratt et al., 2008 |
| $C_{10}H_{18}O_7S$ | 281.0701 | 1.03 | 0.33 | 6.76 | BD | Riva et al., 2016 |
| $C_5H_{10}O_{11}N_2S$ | 304.9783 | 3.60 | 0.31 | 3.32 | BD | Surratt et al., 2008 |
| $C_5H_9O_{13}N_3S$ | 349.9783 | 5.90 | 0.19 | 2.04 | BD | Ng et al., 2008 |
| $C_{10}H_{18}O_8S$ | 297.0650 | 0.75 | 0.14 | 5.25 | BD | Riva et al., 2016 |
| $C_5H_{11}O_8NS$ | 244.0133 | 1.93 | 0.11 | 1.46 | BD | Nestorowicz et al., 2018 |
| $C_5H_9O_{13}N_3S$ | 349.9783 | 5.49 | 0.02 | 0.17 | BD | Ng et al., 2008 |
| $C_5H_9O_{13}N_3S$ | 349.9783 | 5.34 | 0.008 | 0.10 | BD | Ng et al., 2008 |
| $C_5H_{12}O_8S$ | 231.0180 | 0.75 | 0.005 | 0.50 | BD | Riva et al., 2016 |
| $C_{10}H_{20}O_9S$ | 315.0755 | 1.46 | 0.002 | 0.21 | BD | Riva et al., 2016 |

**Table 3.** Isoprene CHO tracer concentrations measured via GC-MS using 24-hour samples between 22/05/2017 and 22/06/2017. 2-MTs is equal to the sum of 2-methylthreitol and 2-methylerythritol and the C5-alkene triols is equal to the sum of cis-2-methyl-1,3,4-trihydroxy-1-butene, 3-methyl-2,3,4-trihydroxy-1-butene and trans-2-methyl-1,3,4-trihydroxy-1-butene.

| Isoprene Tracer | Min (ng m$^{-3}$) | Max (ng m$^{-3}$) | Average (ng m$^{-3}$) |
|---|---|---|---|
| **2-MTs** | 4.55 | 52.67 | 17.29 |
| **MG** | 1.38 | 15.53 | 7.24 |
| **C5-alkene triols** | 0.23 | 1.08 | 0.51 |


**Table 4.** Comparison of concentrations of iSOA tracer concentrations and ratios in previous studies in the Amazon, SE USA and China. *Selected sample not an average concentration.

| Location | Mean Concentration (ng m$^{-3}$) | | | | Ratio low to high NO | | Ratio CHO:CHOS | | Reference |
|---|---|---|---|---|---|---|---|---|---|
| | 2-MT | 2-MT-OS | 2-MG | 2-MG-OS | 2-MT:2-MG | 2-MT-OS:2-MG-OS | 2-MT:2-MT-OS | 2-MG:2-MG-OS | |
| Amazon, Manuas (2016) | 137* | 390* | | | | | 0.35 | | Cui et al., 2018 |
| Amazon, T3 (2014) | | 83(wet)/ 399(dry) | | 0.7(wet)/ 30(dry) | | 118(wet)/ 13(dry) | | | Glasius et al., 2018 |
| SE US, Centreville (2013) | | 217 | | 10.7 | | 20.3 | | | Riva et al., 2019 |
| SE US, Look Rock (2013) | 163.1 | 169.5 | 7.5 | 10 | 21.7 | 17.0 | 0.96 | 0.75 | Budlisuli stiorini et al., |
| SE US, Look Rock (2013) | 861* | 2334* | | | | | 0.37 | | Cui et al., 2018 |
| SE US, Atlanta (2015) | | 1792 | | 53 | | 33.8 | | | Hettiyadura et al., 2019 |
| China Regional, PRD (2008) | 91.5 | 2.2 | 7.7 | 1.4 | 11.9 | 1.57 | 41.6 | 5.51 | He et al., 2018 |
| China Regional, Beijing (2016) | | 5.3 | | 3.6 | | 1.47 | | | Wang et al., 2018 |
| China Rural, NCP (2013) | 44 | | 19.3 | | 2.30 | | | | Li et al., 2018 |
| China Urban, Beijing (2017) | 17.3 | 11.8 | 7.2 | 21.5 | 2.40 | 0.55 | 1.47 | 0.33 | This work |