# Peer review of "Strong anthropogenic control of secondary organic aerosol formation from isoprene in Beijing"

_Atmospheric Chemistry and Physics, 2019_

## Referee Comment (RC1) · Anonymous Referee #1 · 9 Dec 2019

Summary and recommendation:

In this study, Bryant et al. examined the formation of isoprene-derived SOA (iSOA) during summer in Beijing (China) using a large suite of online and offline instruments. In particular, the authors focus on LC-MS data from filter extracts of PM2.5 and the detection, identification, and quantification of isoprene-derived organosulfates (OSs) and nitrooxy-organosulfates (NOSs). They determined an average concentration of iSOA OSs and NOSs of 82.5 ng m–3. Moreover, the authors claim that OS formation depends on a combination of photochemistry and particulate sulfate concentrations, and suggest that iSOA formation is strongly controlled by anthropogenic emissions in Beijing.

The authors acquired an impressive dataset with state-of-the-art instruments during

their field study. Moreover, molecular-level identification and quantification of SOA constituents, such as OSs and NOSs, is challenging, yet highly desirable. However, I see several major weaknesses in the measurement approach, the data analysis and interpretation of the results, which need to be addressed before I can recommend the publication of the manuscript (as detailed below).

Major comments:

1) I have the impression that the main conclusion (which is also the title) of the manuscript is only weakly supported by the data shown in the manuscript. The authors claim that there is a "strong anthropogenic control of SOA formation from isoprene in Beijing". However, only one figure (i.e., Figure 5) is actually showing a weak correlation of OSs to the product of ozone and sulfate concentrations, which can be considered to some extent as a metric for anthropogenic influences. Nonetheless, no correlations of total SOA mass to common anthropogenic pollutants is shown or discussed. It should also be noted that particulate sulfate could have been transported to the sampling site over longer distances, and thus, is not a direct anthropogenic emission factor.

2) In general, the discussion of the data remains quite superficial in several sections, which might also be one of the reasons for my comment above. For example, Fig. 1, which displays typical anthropogenic pollutants, is barely discussed in the results section (i.e. only four sentences, P8L315–318). This is similar for several other figures in the manuscript. Moreover, the authors should reconsider the order and layout of the figures, as the reader is often forced to jump between figures or to find essential data in the SI (e.g., Fig. S10).

3a) One of my most pressing concerns is the LC-MS method used for filter extract analysis and the selection of reversed-phase LC for the separation and detection of isoprene-derived OSs. As can clearly be seen from Table 1, the OSs are not retained on the column and probably elute together with inorganic salts such as sulfate. For example, 15 of the 31 identified species have a retention time of <0.8 min. For these com-

pounds, it is actually impossible to exclude ionization artefacts, i.e., the corresponding compounds are formed from inorganic sulfate and organic compounds during electrospray ionization. It would be necessary to give at least the dead time of the LC system, to make the reader aware of such potential artifacts.

3b) Moreover, the authors state that potential isoprene OSs were "identified" based on previous studies and then searched for in the LC-MS data (cf. P5L204f). However, even with high mass accuracy and resolving power, it is not possible to identify compounds without any reference standards. I can imagine that there were actually several isomers detectable for each of the "identified" OSs in Table 1 (actually, the authors even admit the presence of isomeric species in the following section 2.4). How did the authors decide which retention time to take for each of the OSs from the literature?

4) The authors claim that they account for matrix effects by calibrating the LC-MS through a standard addition approach. However, in my opinion, this approach is far from convincing, as only 6 of the 132 filters (i.e., 4.5%) of all filters were actually investigated. Moreover, it seems quite unreasonable to me to take the calibration curve of 2-MG-OS for all other detected OSs. As shown in Fig. S2, the slope of 2-MT-OS is about 4x larger than for 2-MG-OS. Eventually, such differences in sensitivity might lead to a large overestimation of OSs concentrations. This has to be discussed in more detail. Merely stating that the measurement uncertainty is 60% is not sufficient. How did the authors actually calculate this uncertainty?

Specific comments:

1) As one of the main aspects of the manuscript is the quantification of OSs and NOSs, I was wondering if the introduction may benefit from some more information on the general abundance of OSs also from other precursors such as monoterpenes and/or alkanes (e.g., Wang et al. JGR 2017, doi: 10.1002/2017JD026930; Brüggemann Environ. Chem. 2019, doi: 10.1071/EN19089; Riva et al. ACP 2016, doi:10.5194/acp-16-11001-2016)

[Figure]

2) P5L174: As the samples were only extracted using H2O, it is important to state that only the water-soluble fraction of the filter samples was analyzed.

3) P5L175: Did the authors account for potential artifacts from sonication of the filter samples, for example, as shown by Mutzel et al. (Atmos. Environ. 2013, doi:10.1016/j.atmosenv.2012.11.012)

4) P5L191: What is "full scan MS2"? Do the authors mean full scan combined with data-dependent fragmentation (i.e., full scan-ddMS2)? If so, how many different ion species were fragmented after each full scan? And how were they selected? Moreover, the authors should state at which resolution the mass spectra were obtained, as this will also determine the chromatographic time resolution of the MS method. Besides, it also remains unclear if and how many replicate injections were performed for each sample.

5) P6L216: Here, it is necessary to explain what standards were used for the calibration curves. Without this knowledge, it is very confusing for the reader to follow this discussion.

6) Section 2.5: The filter extraction method for the HILIC-MS analysis is very different from the LC-MS method. In this case, pure methanol was used for extraction. Did the authors investigate how this affects the presence of certain OSs and NOSs?

7) P8L300: It is stated that a SIFT-MS was used to measure isoprene, however, there are no data shown from these measurements. The authors merely claim that there is a good correlation to the DC-GC measurements (P8L321)

8) Fig. 1: I would recommend avoiding interpolation over periods of missing data. Again, as mentioned above, there is almost no discussion of the data.

9) P8L328: Give values for k_ox in the main text and not only in the caption of the figure.

10) Fig. 3a: I would recommend giving the loss rate in ppb h-1, as this would help

the reader to compare the numbers to the actual measurements. Moreover, you could try to estimate a lower limit for isoprene emissions here and compare it to previous studies.

11) P9L349 / Fig. S2-S4: The calibration curves are not of high quality. Especially, for the lower concentrations the data seem to deviate quite strongly from the determined fits (Fig. S2). It would be nice to see errorbars for the variability / standard deviation of the data. Why is the standard addition only shown for one filter sample each? You could normalize the data and show how the slopes of the calibration curves compare to the external calibration for the two compounds.

12) P9L354 / Table S1 and S2: How did the authors select these filter samples? There is no further information given on how these filters were selected and whether they are representative for the campaign period. Moreover, Tables S1 and S2 are very difficult to comprehend.

13) P10L372: Where do these scaling factors come from?

14) P10L374: How did the authors calculate an uncertainty of 60% for their concentrations? If there is such a high uncertainty, it should clearly be stated also in Table 1.

15) Fig. 4: It is misleading to represent the concentrations of the quantified compounds by single data points and to interpolate in between. Better display concentration steps with corresponding lengths to the sampling time.

16) P10L394: The reader has to believe the authors that the three-hourly data are consistent with the hourly samples. No evidence is shown for this assertion.

17) P10L398–P11L413: It would be helpful for the reader to see all the data discussed in this section in one figure. Furthermore, it would be much more convincing to see correlation plots instead of time series (also in other parts of the manuscript).

18) Fig S7 / P11L416: It is obvious that especially OSs with similar retention times

(i.e., 0.71–0.73 min) strongly correlate with each other. This is a strong indication that ionization artifacts produce these species (e.g., fragments, adducts, complexes, etc.), as they are all ionized at the same time in the ESI source. The authors should discuss and clearly state this in the manuscript.

19) P11L421: Any references which support this hypothesis?

20) Fig. S8: The figure has a bad quality and typos in the title. Moreover, the caption is not sufficient to understand the content of the figure.

21) P11L445: Do the authors mean regional background sites? (also at P12L450)

22) Fig. S10 / P12L463: Why is the Figure in the SI? It is an essential part of your data. Any hypothesis why the concentrations are higher at night?

23) Fig. 6: It would be beneficial to see a comparison to total PM or OM mass concentrations in the figure.

24) P13L526: There are no corresponding data shown to this discussion.

25) P14L546: So far, there was no clear discussion of NOx levels and how they affect iSOA formation in Beijing.

Technical comments:

1) P5L184: What is "UPLC-MS2"? I think this term is very misleading here. If I understood the methods section correctly, the authors use full scan data for the quantification of the target analytes.

2) P9L341: The wording "high throughput screening" is quite exaggerated. The method described here is more like a standard data processing method of LC-MS data.

3) P9L355: I do not understand what the sentence "50 $\mu$L of filter sample extract and ..." should tell the reader here.

4) P9L367: Use the notation "Table S1 and S2" to avoid confusion

5) P10L389: I do not understand what this sentence should tell us here. Is this an important comparison to AMS data?

6) P14L540: Instead of "heterogeneous products", better use "products from heterogeneous reactions".

---

## Referee Comment (RC2) · Anonymous Referee #2 · 11 Dec 2019

The paper by Bryant et al. entitled as 'Strong anthropogenic control of secondary organic aerosol formation from isoprene in Beijing' shows the observational data for isoprene-derived organic aerosol (iSOA), especially focusing on orgnaosulfate. Although observational data of iSOA in Beijing itself might be unique, the reviewer believes that the manuscript needs to be re-structured and more carefully prepared to meet the publication criteria for ACP. Discussion of the data could have been deeper and more convincing if the authors have analyzed the data in more detail and carefully. There were numerous errors in the reference list. Some technical descriptions were not clear. I can understand that the authors have put significant effort to obtain this dataset. So, the manuscript should have been prepared more carefully.

Major comments Figures 3 and S1. L331 'OH chemistry is still an important loss route

at night (>30 %) owing to night-time OH sources, such as the ozonolysis of alkenes.' The authors estimated that the contribution of OH radicals on the loss process of isoprene is more than 50% even during nighttime, as the concentration of OH radical is close to 2 × 106 molecules cm-3 even at around the midnight. However, this concentration seems to be higher than OH concentrations in other urban areas during nighttime (Kanaya et al., 2007;Heard et al., 2004). Although I am not familiar with other recent field campaigns for OH radicals in Beijing, a recent modeling result also indicated that OH radical concentration in Beijing should be orders of magnitudes lower than what the authors have reported (Tan et al., 2019). If the night-time OH sources in Beijing is so important for isoprene oxidation chemistry, the potential sources need to be discussed in more detail by employing observation data or by appropriately citing necessary publications.

L421 'The correlation of [O3] [pSO4] '/Figure 5 It seems to me that the authors have assumed that the formation of organosulfate is approximated by the second order reaction with a constant reaction rate. However, it is a multiphase chemistry. In addition, the importance of particle phase acidity and phase are known to influence the rate of such chemical reactions. So, I am not sure if it is a good idea to assume that the rate constant is a fixed value. Figure 4 indicates that the concentrations of all the organosulfates are high during pollution episodes. Although the authors did not provide time-series data for sulfate concentration, I speculate that it must have also been high during these time periods. Considering that the ozone concentration was less variable than particle phase species (Figure 1), I suspect that the correlation shown in Figure 5 is dominantly driven by accumulation of particle phase species (i.e., sulfate and organosulfate) during pollution episodes, although there must have also been in-situ production of these species. The discussion will need to be thoroughly revised, including reconsideration of the metric.

Other comments L300 'OH, HO2 and RO2 concentrations were measured using Fluorescence Assay by Gas Expansion (FAGE) and NO3 concentrations were measured

using Broadband cavity enhanced absorption spectrometry (Zhou et al.,2018).' I checked Zhou et al. (2018); however, I was unable to find the corresponding information (maybe, I missed)?

L321 'There was strong a correlation between the isoprene mixing ratio measured at 8 m by the DC-GC and at 102 m using the SIFT-MS (R2 = 0.77). The SIFT-MS measurements were therefore used to investigate the correlation with iSOA tracers when no DC-GC data was available.' Being correlated does not mean that they are quantitatively the same. Further detailed descriptions would be needed.

Section 3.3 The reviewer believes that the content is more suitable for the experimental section.

L402: 'Zero-dimensional box modelling indicates on some days up to 35 % of the isoprene-derived RO2 radicals can react with HO2 in the afternoon (Newland et al., 2019).' As Newland et al. (2019) has not been published yet, I am unable to evaluate the validity of this description.

Acknowledgement It seems that at least one of the authors is also included in the acknowledgement.

References There are numerous errors. For example, names of some authors are duplicated. Information about issues/page numbers are unavailable for some references. Abbreviations for some journals are not following the standard style. Titles of some papers are missing. Some references are appearing twice (i.e., duplicate). The authors will need to carefully check the reference list again.

Figure 1 It seems that all the data points are connected by line, even if some data are missing. I suggest to change it so that the readers can tell for which time period the data is/is not available.

Figure 3 (caption) What is the unit for the reaction rate constants?

References: Heard, D. E., Carpenter, L. J., Creasey, D. J., Hopkins, J. R., Lee, J.

D., Lewis, A. C., Pilling, M. J., Seakins, P. W., Carslaw, N., and Emmerson, K. M.: High levels of the hydroxyl radical in the winter urban troposphere, Geophys. Res. Lett., 31, 10.1029/2004GL020544, 2004. Kanaya, Y., Cao, R., Akimoto, H., Fukuda, M., Komazaki, Y., Yokouchi, Y., Koike, M., Tanimoto, H., Takegawa, N., and Kondo, Y.: Urban photochemistry in central Tokyo: 1. Observed and modeled OH and HO2 radical concentrations during the winter and summer of 2004, Journal of Geophysical Research: Atmospheres, 112, 10.1029/2007JD008670, 2007. Tan, Z., Lu, K., Jiang, M., Su, R., Wang, H., Lou, S., Fu, Q., Zhai, C., Tan, Q., Yue, D., Chen, D., Wang, Z., Xie, S., Zeng, L., and Zhang, Y.: Daytime atmospheric oxidation capacity in four Chinese megacities during the photochemically polluted season: a case study based on box model simulation, Atmos. Chem. Phys., 19, 3493-3513, 10.5194/acp-19-3493-2019, 2019.

---

## Referee Comment (RC3) · Anonymous Referee #3 · 17 Dec 2019

In this work, the authors have quantified a number of organic tracers, primarily organosulfates or nitroxyorganosulfates in order to better the contribution of isoprene-derived secondary organic aerosol (iSOA) to organic aerosol carbon in the atmosphere. They found that iSOA formation in urban Beijing is strongly controlled by anthropogenic emissions. This work provides new valuable field data to better understand the sources of ambient aerosols in an urban polluted environment. I support the publication of this work in ACP and have some comments for the authors' consideration.

Comments Line 43, "The coelution of the inorganic ions in the extracts caused matrix effects that impacted two authentic standards differently." This is very good finding. However, the authors do not further elaborate this point here. What is the potential

significance of matrix effects on the quantification of the species in this work?

Line 51, "These results indicate for the first time that iSOA formation in urban Beijing is strongly controlled by anthropogenic emissions and results in extensive conversion to heterogeneous OS products". Could the authors further elaborate the correlation between the formation of iSOA controlled by the emissions of anthropogenic pollutants and formation of OSs in a more quantitative way? How significance or any numbers based on their field observations and data?

Line 336, "This indicates that there are significant local emissions of isoprene impacting the measurement site and therefore a high potential for the formation of iSOA in this urban environment." Could the authors further comment whether the sampling site be a representative site for the typical urban environment in most parts of Beijing? Could the observations and results presented in this work largely reflect the title of this paper "Strong anthropogenic control of secondary organic aerosol formation from isoprene in Beijing"?

Line 342, "The full list of iSOA tracers, along with their measured m/z and molecular formula is shown in Table 1, ordered by descending average concentration (weighted by filter sampling time and reported in ng m-3) during the campaign" What are the uncertainties of the reported concentrations? Please present the uncertainties.

Line 360, "A strong matrix effect was observed for the 2-MT-OS, with the concentration measured by standard addition calibration 8.6 to 10 times higher than when using the external calibration carried out on the same day." This is an important finding. Why? Could the authors explain these results?

Line 404, "Zero-dimensional box modelling indicates on some days up to 35 % of the isoprene-derived RO2 radicals can react with HO2 in the afternoon (Newland et al., 2019)." Please kindly note that this paper is under review or has been accepted for publication.

[Figure]

Line 425, "Therefore, these spikes in 2-MT-OS could be a result of either higher 2-MT-OS in regional aerosol transported to the site or a high isoprene emission source to the south west of the site (i.e. producing IEPOX locally) that then reacts with increased regional sulfate pollution." Any field evidences or model predictions show that the IEPOX can be effectively produced locally? What are the contributions from the regional transport? The effects of anthropogenic emission on iSOA formation observed in this work are local, regional or a combination of both effects.

Line 450, "The ratio of 2-MT-OS:2-MG-OS observed in Beijing is compared to previous studies in Table 2 and is considerably lower than measurements taken in a range of isoprene dominated environments (South East US, 2-MT-OS:2-MG-OS = 17, Budisolistiorini et al., 2015.; Amazon, 2-MTOS:2-MG-OS = 13-118, Glasius et al., (2018).; Atlanta, 2-MT-OS:2-MG-OS = 33, Hettiyadura et al., (2019)) reflecting the strong impact of urban NO emission on iSOA formation." I agree with this argument. However, how could we use this ratio to quantify the effect of NO emission on iSOA formation or OS formation in different regions?

Line 505, "Some of the NOS observed peaked in the daytime and some were enhanced at night. In total they had a mean concentration of 24 ng m-3 during the campaign. The sources and formation of these species will be discussed in a separate publication." I am okay with this. However, could the authors briefly elaborate how the detection of NOS help to better understand the effect of anthropogenic emission on the iSOA formation in this work?

---

## Author Comment (AC1) · 28 Feb 2020

**We thank the reviewers for their helpful comments. The comments and suggestions have been addressed accordingly. Thank you for your time and effort. The replies to each comment are shown in blue.**

**Reviewer 1:**

In this study, Bryant et al. examined the formation of isoprene-derived SOA (iSOA) during summer in Beijing (China) using a large suite of online and offline instruments. In particular, the authors focus on LC-MS data from filter extracts of PM2.5 and the detection, identification, and quantification of isoprene-derived organosulfates (OSs) and nitrooxy-organosulfates (NOSs). They determined an average concentration of iSOA OSs and NOSs of 82.5 ng m–3. Moreover, the authors claim that OS formation depends on a combination of photochemistry and particulate sulfate concentrations, and suggest that iSOA formation is strongly controlled by anthropogenic emissions in Beijing.

The authors acquired an impressive dataset with state-of-the-art instruments during their field study. Moreover, molecular-level identification and quantification of SOA constituents, such as OSs and NOSs, is challenging, yet highly desirable. However, I see several major weaknesses in the measurement approach, the data analysis and interpretation of the results, which need to be addressed before I can recommend the publication of the manuscript (as detailed below).

**Major comments:**

1) I have the impression that the main conclusion (which is also the title) of the manuscript is only weakly supported by the data shown in the manuscript. The authors claim that there is a "strong anthropogenic control of SOA formation from isoprene in Beijing". However, only one figure (i.e., Figure 5) is actually showing a weak correlation of OSs to the product of ozone and sulfate concentrations, which can be considered to some extent as a metric for anthropogenic influences. Nonetheless, no correlations of total SOA mass to common anthropogenic pollutants is shown or discussed. It should also be noted that particulate sulfate could have been transported to the sampling site over longer distances, and thus, is not a direct anthropogenic emission factor.

There was a similar comment from all three reviewers and so we have added an additional section on the gas phase data (see reply to comment 2) to show the concentrations and trends of anthropogenic tracers in Beijing. We have also added a PollutionRose plot (new Figure 2) of particulate sulfate concentrations, which shows a strong anthropogenic source of particulate sulfate to the south of Beijing which contains a large industrial region. While the sulfate is likely formed outside the city, it is still an anthropogenic tracer. We also discussed in the paper the effect of NO (an anthropogenic tracer) on the products formed and have added additional clarification of the role of biogenic-anthropogenic interactions in the formation mechanisms of the products. Finally, we have shown the moderate to strong relationship with the product of sulfate and ozone, which are both secondary pollutants formed from anthropogenic emissions. We have moved Figure S10 into the main manuscript (new Figure 5) which shows the degree of correlation for all isoprene OS SOA tracers against each other and with a range of anthropogenic tracers.

We have added the following text to address this comment:

"The isoprene SOA tracers identified in this study are correlated towards themselves as well as common anthropogenic tracers in a corplot (Openair, R), shown in Figure 5. The corplot highlights the correlations of the iSOA tracers to each other as well as the moderate to strong correlations towards some of the anthropogenic pollutants as discussed in further sections."

"As such, this tracer is the result of a direct biogenic-anthropogenic interaction."

[Figure]

Frequency of counts by wind direction (%)

"**Figure 2.** PollutionRose plot (Openair) of particle sulfate measured by AMS, during the sampling period."

2) In general, the discussion of the data remains quite superficial in several sections, which might also be one of the reasons for my comment above. For example, Fig. 1, which displays typical anthropogenic pollutants, is barely discussed in the results section (i.e. only four sentences, P8L315–318). This is similar for several other figures in the manuscript. Moreover, the authors should reconsider the order and layout of the figures, as the reader is often forced to jump between figures or to find essential data in the SI (e.g., Fig. S10).

There have been a number of publications concerning the gas phase data presented and so we did not want to repeat that discussion in length here. However, in response to reviewer 1 and 2, we have added a small section discussing the concentrations and trends of the data in Figure 1. We have also moved figure S10 to the main paper in response to the reviewers paper.
We have added the following text to address this comment::

"**3.2 Anthropogenic tracers**

A range of gas phase anthropogenic tracers were measured during the campaign as discussed in Shi et al., 2019. Figure 1 shows the time series of NO, $NO_2$, $O_3$ and particulate sulfate during the part of the campaign analysed in this study. Table 1 shows the average, maximum and minimum concentrations for these anthropogenic pollutants. NO mixing ratios ranged from less than 0.1 ppbv to 104 ppbv, and a mean concentration during the filter sampling period of 5.1 ppb. The highest concentrations generally occurred in the morning 04:00-07:00 and steadily decreased during the day. On some days, the mixing ratio of NO was very low in the afternoon, as a result of reaction with ozone and other unknown sinks (Newland et al., 2020). The mean mixing ratio of $NO_2$ was 22.3 ppbV, much higher than NO, with a range of 3.7 to 95 ppbv. $NO_2$ peaked between 06:00-07:00 and decreased to a minima at 14:00 and then steady increased until about 20:00. High afternoon concentrations of $O_3$ (>80 ppb) were found on most days, with a maximum observed mixing ratio of 182 ppbv. Night time $O_3$ levels were much lower due to reduced photochemistry and reaction with NO, although on some nights $O_3$ levels were maintained above 40 ppbv, as shown in Figure 1. Particulate sulfate concentrations, measured by AMS are also shown in figure 1. Sulfate ranged from 0.7 to 21.7 µg m⁻³, with an average of 5.5 µg m⁻³. The time series shows a number of periods of high sulfate concentrations and these generally matched periods of increased $PM_{2.5}$ (see Figure 9). Figure 2 shows the wind direction dependent concentrations of particulate sulfate for the sampling period in a pollutionRose plot (Openair package, R). There is a strong source of sulfate from the south of the sampling site, which is enhanced under the highest wind speeds. Previous studies have shown a strong source of pollution from the south west of Beijing, which is where many industrial factories are located (Wang et al., 2005).

3a) One of my most pressing concerns is the LC-MS method used for filter extract analysis and the selection of reversed-phase LC for the separation and detection of isoprene-derived OSs. As can clearly be seen from Table 1, the OSs are not retained on the column and probably elute together with inorganic salts such as sulfate. For example, 15 of the 31 identified species have a retention time of <0.8 min. For these compounds, it is actually impossible to exclude ionization artefacts, i.e., the corresponding compounds are formed from inorganic sulfate and organic compounds during electrospray ionization. It would be necessary to give at least the dead time of the LC system, to make the reader aware of such potential artefacts.

Sulfate and nitrate ions elute in the dead time of the column at 0.67 min. Thus these inorganic ions are offset from the OS species, although there is still some co-elution. We analysed a subset of the filters using normal phase HILIC separation and found a strong correlation between the sum of the 2MT-OS isomers and the single mixed isomer peak found using the reverse phase analysis method ($R^2 = 0.85$). Considering the two methods extract into different solvents (water v methanol) this suggests that there is no significant formation of OS species in the ion source. To test for artefact formation of OS in the sources, we analysed a sample containing a large excess of ammonium sulfate with 2-methyltetrol and

2-methylglyceric acid authentic standards. No 2-MG-OS formation was observed and <0.5% conversion was seen for the 2-MT. The combination of both pieces of evidence, excludes the formation of OS in the source. In addition, if OS species form from reactions between oxidised species and sulfate in the ion source, then all previous direct infusion studies would also suffer from this interference. We have added text to clarify this and included this new test.

Added text:
"Due to the wide range of compounds studied, poor retention was observed for some species (RT < 0.8min). These species closely eluted to the dead time of the column where inorganic sulfate ions eluted (0.67 min). To check for ionisation artefacts, a aqueous solution containing 20 ppm ammonium sulfate, 1ppm 2-methytetrol and 1ppm 2-methylglyceric acid was ran under the same conditions as the filter samples to check for organosulfate formation (2-MT-OS and 2-MG-OS respectively). No MG-OS formation was observed and <0.5% conversion was seen for the 2-MT."
"The sum of peak areas from the 2-MT-OS isomers measured by HILIC and the quantified 2-MT-OS (sum of isomers) measured via UPLC/ESI-HR-MS were compared and showed a high degree of correlation ($R^2 = 0.84$), even though the two methods used different solvents. The agreement indicates that the UPLC/ESI-HR-MS method captures the sum of the isomers and there is no evidence of ion source induced artefacts."

3b) Moreover, the authors state that potential isoprene OSs were "identified" based on previous studies and then searched for in the LC-MS data (cf. P5L204f). However, even with high mass accuracy and resolving power, it is not possible to identify compounds without any reference standards. I can imagine that there were actually several isomers detectable for each of the "identified" OSs in Table 1 (actually, the authors even admit
the presence of isomeric species in the following section 2.4). How did the authors decide which retention time to take for each of the OSs from the literature?

We thank the reviewer for highlighting our confusing use of the word identified here. For the majority of the OS species in this study, the separation was not good enough to separate isomers and a single peak was observed in the extracted chromatograms for each $m/z$. The $MS^2$ spectra for each peak was checked for characteristic OS fragmentation patterns. There were only two available authentic standards and so these compounds have been identified in this work (although these still include a number of co-eluting isomers). It is possible that some of the iSOA OS observed here may have other sources as we highlight in the paper. For the NOS, the individual isomers could be resolved and are presented in the paper. We have replaced the confusing text.
Following text was changed:
"Using previously observed iSOA products from literature, extracted ion chromatograms were plotted for each *m/z* value from a small subset of ambient samples and the retention time (RT) of observed species/isomer were obtained. For most of the OS species in this study the separation was not good enough to see individual isomers and only one peak was observed, which was added to the library. For the NOS species, individual isomers could be resolved, and each isomer was added to the library based on its retention time."
L222 : "Tracefinder extracted the OS/NOS tracer peak areas from each ambient sample chromatogram using the library based on RT and accurate mass."

4) The authors claim that they account for matrix effects by calibrating the LC-MS through a standard addition approach. However, in my opinion, this approach is far from convincing, as only 6 of the 132 filters (i.e., 4.5%) of all filters were actually investigated. Moreover, it seems quite unreasonable to me to take the calibration curve of 2-MG-OS for all other detected OSs. As shown in Fig. S2, the slope of 2-MT-OS is about 4x larger than for 2-MG-OS. Eventually, such differences in sensitivity might lead to a large overestimation of OSs concentrations. This has to be discussed in more detail. Merely stating that the measurement uncertainty is 60% is not sufficient. How did the authors actually calculate this uncertainty?

We disagree with the reviewers comments on using the 2-MG-OS to represent the other iSOA compounds. The 2-MT-OS calibration line using standard addition is only 2.33 times higher than the 2-MG-OS external calibration, compared to around 10 times higher using the 2-MT-OS external calibration. If we used the external calibrations we would have significantly **underestimated** the concentration of the 2-MT-OS species. It is true, that using proxys to calibrate the other iSOA could result in issues during the calibration. However, this careful analysis was carried out to minimise the effects of using just a single external calibration to estimate concentrations. We have added additional text to discuss the potential for underestimating the concentrations without taking matrix effects into account. The 60 % uncertainty was used to account for the difference in the observed correction factors used when correcting for matrix effects.
added text:" this uncertainty was calculated to account for the difference in the measured correction factors used when correcting for the matrix effects."
added text: " Further work is needed to fully understand the reasons. Without these additional standard addition calibrations, the iSOA concentrations would have been largely underestimated."

Specific comments:
1) As one of the main aspects of the manuscript is the quantification of OSs and NOSs, I was wondering if the introduction may benefit from some more information on the general abundance of OSs also from other precursors such as monoterpenes and/or alkanes (e.g., Wang et al. JGR 2017, doi: 10.1002/2017JD026930; Brüggemann Environ. Chem. 2019, doi: 10.1071/EN19089; Riva et al. ACP 2016, doi:10.5194/acp-16-11001-2016)
The following text has been added:
"SOA formed from anthropogenic and other biogenic (monoterpenes and sesquiterpenes) sources have also been studied. Thousands of organic species including hundreds of OS and nitroxy OS (NOS) species have been identified studies from a range of precursors using UPLC/ESI-HR-MS from ambient aerosol samples (Wang et al., 2016, Wang et al., 2017). Brüggemann et al. (2019) quantified with authentic standards both monoterpene OS (MT-OS) and sesquiterpene OS (SQT-OS) species in Melpitz, Germany and Wangdu, China. They found median daytime concentrations for Melpitz and Wangdu for 52 MT-OS species of 12.15 ng m$^{-3}$ and 38.19 ng m$^{-3}$ respectively. For the 5 SQT-OS species, median concentrations were 0.3 ng m$^{-3}$ and 3.90 ng m$^{-3}$ for daytime concentrations respectively, much lower than the iSOA OS species quantified in this study. Riva et al. (2016) identified OS species from the photo-oxidation of $C_{10} - C_{12}$ alkanes, which were then characterised in ambient aerosol samples collected in Lahore, Pakistan and Pasadena, CA, USA. High concentrations of OS species were identified in Lahore, with the largest observed concentration arising from a cycodecane OS species ($C_{10}H_{16}O_7S$) with a concentration of 35.93 ng m$^{-3}$. "

2) P5L174: As the samples were only extracted using H2O, it is important to state that only the water-soluble fraction of the filter samples was analyzed.

The following text has been added: "The water-soluble fraction of the filter samples were analysed"

3) P5L175: Did the authors account for potential artifacts from sonication of the filter samples, for example, as shown by Mutzel et al. (Atmos. Environ. 2013, doi:10.1016/j.atmosenv.2012.11.012)

Additional experiments comparing sonication and an orbital shaker found no appreciable difference in the concentrations of iSOA tracers analysed in this study.

Following text has been added:

 "A small subset (3) of the filter samples were also extracted via orbital shaker and no appreciable difference was found in the concentrations of the iSOA tracers compared to sonication."

4) P5L191: What is "full scan MS2"? Do the authors mean full scan combined with data-dependent fragmentation (i.e., full scan-ddMS2)? If so, how many different ion species were fragmented after each full scan? And how were they selected? Moreover, the authors should state at which resolution the mass spectra were obtained, as this will also determine the chromatographic time resolution of the MS method. Besides,
it also remains unclear if and how many replicate injections were performed for each sample.

The following text has been added:

"analysed using  UPLC-full scan-ddMS$^2$ "
", with a resolution of 70,000"
", each filter sample was run only once."
" The number of most abundant precursors for MS$^2$ fragmentation per scan was set to 10"

5) P6L216: Here, it is necessary to explain what standards were used for the calibration curves. Without this knowledge, it is very confusing for the reader to follow this discussion.

This section is simply discussing how the data was handled, and therefore doesn't go into detail surrounding how the calibrations were achieved. The quantification including standards are discussed in section 3.3. We have added a sentence to direct the reader to the later section.

Following text has been added:
"(as discussed in section 3.3)"

6) Section 2.5: The filter extraction method for the HILIC-MS analysis is very different from the LC-MS method. In this case, pure methanol was used for extraction. Did the authors investigate how this affects the presence of certain OSs and NOSs?

This was not carried out as only a small subset of filters were analysed and thus whether extraction in methanol favours certain species over other has not been determined. However, the 2-MT-OS data obtained using both methods show a strong correlation as mentioned above, but unfortunately the HILIC method was not calibrated at the time.

7) P8L300: It is stated that a SIFT-MS was used to measure isoprene, however, there are no data shown from these measurements. The authors merely claim that there is a good correlation to the DC-GC measurements (P8L321)

The DC-GC Isoprene diurnal profile is shown in figure 3 and the full time series is shown in figure 1. We did not show the sift data since it was taken at a height of 100 m up the tower. We only used the SIFT data to compare the correlation of the iSOA species with both techniques, as they cover slightly different time periods due to instrument maintenance and down-time.

8) Fig. 1: I would recommend avoiding interpolation over periods of missing data.
Again, as mentioned above, there is almost no discussion of the data.

This has been addressed with a new section 3.2. Interpolation across periods of missing data removed.

[Figure]

9) P8L328: Give values for k_ox in the main text and not only in the caption of the figure.
Following text has been added:
L343: "The IUPAC rate constants that were used in the calculation for $NO_3$, $O_3$ and OH were $7 \times 10^{-13}$, $1.27 \times 10^{17}$, $1 \times 10^{10}$ $cm^3$ molecule$^{-1}$ s$^{-1}$ respectively."
10) Fig. 3a: I would recommend giving the loss rate in ppb h-1, as this would help the reader to compare the numbers to the actual measurements. Moreover, you could try to estimate a lower limit for isoprene emissions here and compare it to previous studies.

We have added a figure showing the values in ppb hr-1. We cannot estimate the isoprene emission as we don't know the reaction time between emission and measurement and since this is a biogenic emission, photochemical age calculations are unlikely to give useful information.

[Figure]

[Figure]

**Figure 4.** (A) Diurnal loss rate of isoprene calculated using measured average diurnal profiles of isoprene, OH, $NO_3$ and ozone. (B) Average diurnal of the percentage loss of isoprene from reactions with OH, $O_3$ and $NO_3$ radicals. The IUPAC rate constants used for the calculations are as follows, $NO_3$: $7 \times 10^{-13}$, $O_3$: $1.27 \times 10^{17}$, OH: $1 \times 10^{10}$ $cm^3$ $molecule^{-1}$ $s^{-1}$ (Atkinson et al., 2006).

11) P9L349 / Fig. S2-S4: The calibration curves are not of high quality. Especially, for
the lower concentrations the data seem to deviate quite strongly from the determined fits (Fig. S2). It would be nice to see errorbars for the variability / standard deviation of the data. Why is the standard addition only shown for one filter sample each? You could normalize the data and show how the slopes of the calibration curves compare to the external calibration for the two compounds.

There is only a small deviation at the lower concentrations, with both calibrations showing a good linear fit. Error bars have been added to the calibration curves. Only one standard addition per filter sample was carried out due to having a limited amount of sample. We did present all of the standard addition sample data (i.e. the comparison of slopes) in the Table S1 and S2 but from the reviewers comment this is not clear to the reader. The additional text we have added in response to comment 12
clarifies this.

[Figure]

12) P9L354 / Table S1 and S2: How did the authors select these filter samples? There is no further information given on how these filters were selected and whether they are
representative for the campaign period. Moreover, Tables S1 and S2 are very difficult to comprehend.

This sentence in the text already describes how these filters were selected:

"Five-point standard addition calibrations were run on 6 different filter extracts, covering both day and nighttime samples, during periods of both high and low concentrations of iSOA species."

An additional paragraph has been added to improve understanding of tables S1 and S2.

"Table S1 shows the concentration of 2-MT-OS in three filter sample extracts (144, 204, 208) calculated via standard addition of 2-MT-OS to the filter sample extract and via external calibrations using both 2-MT-OS and 2-MG-OS. The ratio of the standard addition to the external calibrations then gives an estimate of the under or overestimate the external calibrations make to calculating the concentration of 2-MT-OS in the samples. Both the external calibrations would lead to an underestimation of concentration of 2-MT-OS in the filter samples. 2-MG-OS provided a closer quantification of 2-MT-OS in the samples, with an average factor of 2.3 underestimation, while the 2-MT-OS external calibration gives a sample concentration a factor of 10 lower than the standard addition determined concentration."

13) P10L372: Where do these scaling factors come from?
This has been addressed in response to comment 12.

14) P10L374: How did the authors calculate an uncertainty of 60% for their concentrations? If there is such a high uncertainty, it should clearly be stated also in Table 1.

This is addressed above.  We do not feel it is necessary to add this to the table.

15) Fig. 4: It is misleading to represent the concentrations of the quantified compounds by single data points and to interpolate in between. Better display concentration steps
with corresponding lengths to the sampling time.

We have modified the figures to show the start and end time of the samples but kept a data point at the mid sample time and the interpolation between points.  Using simple bars made the plot very difficult to read, especially as there are over 100 sample points.

[Figure]

16) P10L394: The reader has to believe the authors that the three-hourly data are consistent with the hourly samples. No evidence is shown for this assertion.

Both figures 2 (now 3) and S10 (now 8) show the diurnal variation of 2-MT-OS and 2-MG-OS respectively during the hourly sampling and shows that the 1 hourly is consistent with the 3 hourly data. The hourly filters were taken during the highest pollution event, and so is expected to show higher concentrations.

17) P10L398–P11L413: It would be helpful for the reader to see all the data discussed in this section in one figure. Furthermore, it would be much more convincing to see correlation plots instead of time series (also in other parts of the manuscript).

A Corplot has now been added to the main manuscript to highlight the correlations between all known iSOA species and anthropogenic pollutants. This clearly shows the high correlation of different types of iSOA with each other, with the daytime peaking OS in the top right and the NOS towards the middle of the plot. The gas phase tracers are in the bottom corner. This plot also includes both [SO42-] and [O3][SO42-] and the higher correlations with the latter can be seen.

[Figure]

**Figure 5.** Corplot (Openair) highlighting the correlations between known iSOA tracers and anthropogenic pollutants. The number represents the R correlation between the two species. With redder more elongated circles highlighting a higher correlation.

18) Fig S7 / P11L416: It is obvious that especially OSs with similar retention times(i.e., 0.71–0.73 min) strongly correlate with each other. This is a strong indication that ionization artifacts produce these species (e.g., fragments, adducts, complexes, etc.),as they are all ionized at the same time in the ESI source. The authors should discuss and clearly state this in the manuscript.
See reply to comment 3a.

19) P11L421: Any references which support this hypothesis?
This sort of correlation has not been seen previously as far as we are aware and so no reference can be included.

20) Fig. S8: The figure has a bad quality and typos in the title. Moreover, the caption is not sufficient to understand the content of the figure.

We agree that the labels on the plot have gone wrong during the conversion to pdf. This has been rectified. We have added additional text to the caption.

New figure caption: "**Figure S7:** Extracted ion chromatograms (m/z 215.0) using hydrophobic interaction liquid chromatography (HILIC) of 2-methyltetrol OS (2-MT-OS) isomers, highlighting improved separation of isomers using this technique. Upper: Extract of filter collected on the 28/05/2017 between 11:33 and 14:23. Lower: 10 ppm 2-MT-OS standard (Cui et al., 2018)."

21) P11L445: Do the authors mean regional background sites? (also at P12L450)

This has been addressed.
Text added: "background"

22) Fig. S10 / P12L463: Why is the Figure in the SI? It is an essential part of your data.
Any hypothesis why the concentrations are higher at night?
Moved into the main paper. We are not sure what the second part of this comment refers to?

23) Fig. 6: It would be beneficial to see a comparison to total PM or OM mass concentrations
in the figure.
We have added PM2.5 into an upper panel on figure 6 for comparison.

[Figure]

24) P13L526: There are no corresponding data shown to this discussion.
A table (Table 3) has been added to show the min, max and mean concentrations of the CHO species
measured via GC-MS. The following has been added:

"**Table 3.** Isoprene CHO tracer concentrations measured via GC-MS using 24 hour samples between 22/05/2017
and 22/06/2017. 2-MTs is equal to the sum of 2-methylthreitol and 2-methylerythritol and the C5-alkene triols is
equal to the sum of cis-2-methyl-1,3,4-trihydroxy-1-butene, 3-methyl-2,3,4-trihydroxy-1-butene and trans-2-
methyl-1,3,4-trihydroxy-1-butene."

| Isoprene Tracer | Min (ng m$^{-3}$) | Max (ng m$^{-3}$) | Average (ng m$^{-3}$) |
| --- | --- | --- | --- |
| 2-MTs | 4.55 | 52.67 | 17.29 |
| MG | 1.38 | 15.53 | 7.24 |
| C$_5$-alkene triols | 0.23 | 1.08 | 0.51 |

25) P14L546: So far, there was no clear discussion of NOx levels and how they affect iSOA formation in Beijing.

This has been addressed by the response to comment 2.

Technical comments:

1) P5L184: What is "UPLC-MS2"? I think this term is very misleading here. If I understood the methods section correctly, the authors use full scan data for the quantification of the target analytes.

We do indeed use full scan for the quantification but we also use the MS2 to check that the species fragment to give typical OS fragment ions. This has been added to the text.

Following text was added:

"It should be noted that MS$^2$ was used to check that the iSOA species fragmented to give typical OS fragment ions."

2) P9L341: The wording "high throughput screening" is quite exaggerated. The method described here is more like a standard data processing method of LC-MS data.

The high throughput represents the analysis time, which at 16 min is shorter than previous methods. By building the library of dedicated iSOA species, we have also significantly reduced the analysis time.

3) P9L355: I do not understand what the sentence "50 _L of filter sample extract and : : :" should tell the reader here.

In the pdf version we downloaded this sentence says 50 MICRO(µ) L

4) P9L367: Use the notation "Table S1 and S2" to avoid confusion

This has now been changed.

5) P10L389: I do not understand what this sentence should tell us here. Is this an important comparison to AMS data?

This information shows that under these conditions the AMS cannot replicate the type of isoprene PMF factors analysis that can be done in more isoprene dominated regions.

6) P14L540: Instead of "heterogeneous products", better use "products from heterogeneous reactions".

This has now been changed.

Reviewer 2

The paper by Bryant et al. entitled as 'Strong anthropogenic control of secondary organic aerosol formation from isoprene in Beijing' shows the observational data for isoprene-derived organic aerosol (iSOA), especially focusing on orgnaosulfate. Although observational data of iSOA in Beijing itself might be unique, the reviewer believes that the manuscript needs to be re-structured and more carefully prepared to meet the publication criteria for ACP. Discussion of the data could have been deeper and more convincing if the authors have analyzed the data in more detail and carefully. There were numerous errors in the reference list. Some technical descriptions were not clear. I can understand that the authors have put significant effort to obtain this
dataset. So, the manuscript should have been prepared more carefully.

We thank the reviewer for their helpful comments and suggestions.  However, we do not understand why they think it does not meet "ACP criteria"?  We feel the paper has been written carefully and we have added clarification to the points below.  When large scale field experiments are carried out, there are often numerous groups concentrating their publications on specific areas.  Here we are interested in the isoprene SOA and feel that an extensive discussion of the gas phase data is not needed and can be found in Shi et al., 2019 and other subsequent research papers.  However, we have added a small section to draw out the main conclusions from the auxiliary data as shown in reviewer 1 comment 2 and the new section 3.2

We could only find a few errors and one duplication in the reference list and these have been addressed.

Major comments

Major comments Figures 3 and S1. L331 'OH chemistry is still an important loss route at night (>30 %) owing to night-time OH sources, such as the ozonolysis of alkenes.' The authors estimated that the contribution of OH radicals on the loss process of isoprene is more than 50% even during nighttime, as the concentration of OH radical is close to 2 _ 106 molecules cm-3 even at around the midnight. However, this concentration seems to be higher than OH concentrations in other urban areas during nighttime (Kanaya et al., 2007;Heard et al., 2004). Although I am not familiar with other recent field campaigns for OH radicals in Beijing, a recent modeling result also indicated that OH radical concentration in Beijing should be orders of magnitudes lower than what the authors have reported (Tan et al., 2019). If the night-time OH sources in Beijing is so important for isoprene oxidation chemistry, the potential sources need to be discussed in more detail by employing observation data or by appropriately citing necessary publications.

Actually, significant OH concentrations have been seen at night for field campaigns performed in both the Beijing and Pearl River Delta areas in China. In Lu at el. (ACP 2014) OH concentrations of up to

$3\times10^6$ at night (higher than those reported for our work at night) were observed and there is a modelling study of two campaigns in that paper which discusses radical sources at night, and it was proposed that these levels can be reconciled if the model includes an additional ROx production process. A further discussion of night-time chemistry is out of scope of the current paper, there are other papers in preparation which will discuss night-time radical chemistry in more detail. We have added Lu et al. as a reference to potential sources of OH and other radicals at night in this type of environment in China."
Lu, K. D., Rohrer, F., Holland, F., Fuchs, H., Brauers, T., Oebel, A., Dlugi, R., Hu, M., Li, X., Lou, S. R., Shao, M., Zhu, T., Wahner, A., Zhang, Y. H., and Hofzumahaus, A.: Nighttime observation and chemistry of HOx in the Pearl River Delta and Beijing in summer 2006, Atmos. Chem. Phys., 14, 4979–4999, https://doi.org/10.5194/acp-14-4979-2014, 2014.

L421 'The correlation of [O3] [pSO4] '/Figure 5 It seems to me that the authors have assumed that the formation of organosulfate is approximated by the second order reaction with a constant reaction rate. However, it is a multiphase chemistry. In addition, the importance of particle phase acidity and phase are known to influence the rate of such chemical reactions. So, I am not sure if it is a good idea to assume that the
rate constant is a fixed value. Figure 4 indicates that the concentrations of all the organosulfates are high during pollution episodes. Although the authors did not provide time-series data for sulfate concentration, I speculate that it must have also been high during these time periods. Considering that the ozone concentration was less variable than particle phase species (Figure 1), I suspect that the correlation shown in Figure 5 is dominantly driven by accumulation of particle phase species (i.e., sulfate and organosulfate) during pollution episodes, although there must have also been insitu production of these species. The discussion will need to be thoroughly revised, including reconsideration of the metric.

We disagree with the reviewers comments here.  The correlation between the OS and particulate sulfate is weaker than the correlation with [O3][SO4] as shown in figure S10 (now the new Figure 5). We are not suggesting there is a direct reaction with a fixed rate constant under all conditions, but at this location ozone is acting as a proxy of the photochemical oxidation of isoprene, leading to the precursors of the OS. Therefore their formation rates are dependent on the production of isoprene oxidation products and the particle phase reaction(s) with sulfate. This relationship may not hold at longer photochemical lifetimes and under different conditions.

Other comments L300 'OH, HO2 and RO2 concentrations were measured using Fluorescence Assay by Gas Expansion (FAGE) and NO3 concentrations were measured using Broadband cavity enhanced absorption spectrometry (Zhou et al.,2018).' I checked Zhou et al. (2018); however, I was unable to find the corresponding information (maybe, I missed)?

This has been addressed. Cited Whalley et al., 2010.

L321 'There was strong a correlation between the isoprene mixing ratio measured at 8 m by the DC-

GC and at 102 m using the SIFT-MS ($R2 = 0.77$). The SIFT-MS measurements were therefore used to investigate the correlation with iSOA tracers when no DC-GC data was available.' Being correlated does not mean that they are quantitatively the same. Further detailed descriptions would be needed.

There was an offset in concentration between the isoprene measured at the ground and at 100 m reflecting the loss rate of isoprene during transport from surface emissions. We originally included this information and felt it was not needed. However, we have added it back in to address the reviewers comment.

Following text has been added:

L348 "The slope of the linear fit between the two data sets was 0.67, indicating a loss of around 30% of the isoprene during transport from the ground to the tower(100m)."

Section 3.3 The reviewer believes that the content is more suitable for the experimental section.

We thank the reviewer for their suggestion. However, we believe that section 3.3 should remain as part of the main text due to identifying the matrix effects as a key result of this study.

L402: 'Zero-dimensional box modelling indicates on some days up to 35 % of the isoprene-derived RO2 radicals can react with HO2 in the afternoon (Newland et al., 2019).' As Newland et al. (2019) has not been published yet, I am unable to evaluate the validity of this description.

Newland et al., 2019 has now been published in ACPD. https://www.atmos-chem-phys-discuss.net/acp-2020-35/

Acknowledgement It seems that at least one of the authors is also included in the acknowledgement.

Removed Yele Sun from Acknowledgements

References There are numerous errors. For example, names of some authors are duplicated. Information about issues/page numbers are unavailable for some references.

Abbreviations for some journals are not following the standard style. Titles of some papers are missing. Some references are appearing twice (i.e., duplicate). The authors will need to carefully check the reference list again.

These have been addressed.

Figure 1 It seems that all the data points are connected by line, even if some data are missing. I suggest to change it so that the readers can tell for which time period the data is/is not available.

Addressed in previous comment to reviewer 1 and now in the new version of Figure 1.

Figure 3 (caption) What is the unit for the reaction rate constants?

Text added: "$cm^3$ molecule$^{-1}$ s$^{-1}$"

**Reviewer 3**

Comments:

In this work, the authors have quantified a number of organic tracers, primarily organosulfates or nitroxyorganosulfates in order to better the contribution of isoprene derived secondary organic aerosol (iSOA) to organic aerosol carbon in the atmosphere. They found that iSOA formation in urban Beijing is strongly controlled by anthropogenic emissions. This work provides new valuable field data to better understand the sources of ambient aerosols in an urban polluted environment. I support the publication of this work in ACP and have some comments for the authors' consideration.

Comments Line 43, "The coelution of the inorganic ions in the extracts caused matrix effects that impacted two authentic standards differently." This is a very good finding. However, the authors do not further elaborate this point here. What is the potential significance of matrix effects on the quantification of the species in this work?

This has been addressed in response to reviewer one, comment 4.

Line 51, "These results indicate for the first time that iSOA formation in urban Beijing is strongly controlled by anthropogenic emissions and results in extensive conversion to heterogeneous OS products". Could the authors further elaborate the correlation between the formation of iSOA controlled by the emissions of anthropogenic pollutants and formation of OSs in a more quantitative way? How significance or any numbers based on their field observations and data?

We have addressed this in our response to reviewer 1, comment 1.

Line 336, "This indicates that there are significant local emissions of isoprene impacting the measurement site and therefore a high potential for the formation of iSOA in this urban environment."

Could the authors further comment whether the sampling site be a representative site for the typical urban environment in most parts of Beijing? Could the observations and results presented in this work largely reflect the title of this paper "Strong anthropogenic control of secondary organic aerosol formation from isoprene in Beijing"?

The site which was located between the fourth and third north ring roads of Beijing is a largely residential area which is typical of Beijing as described in Shi et al. As such, this study is thought to be representative of Beijing as a whole. Beijing has a very high proportion of green space (55%) indicating there is significant coverage of land with plants which could emit isoprene.

Line 342, "The full list of iSOA tracers, along with their measured m/z and molecular formula is shown in Table 1, ordered by descending average concentration (weighted by filter sampling time and reported in ng m-3) during the campaign" What are the uncertainties of the reported concentrations? Please present the uncertainties.

The estimated uncertainties were discussed in section 3.4 as 60 %, accounting for the use of the matrix correction factors.

Line 360, "A strong matrix effect was observed for the 2-MT-OS, with the concentration measured by standard addition calibration 8.6 to 10 times higher than when using the external calibration carried out on the same day." This is an important finding. Why? Could the authors explain these results?

The extracted samples are a complex mixture of different compounds, including a high proportion of inorganic ions that are extracted into water. This is likely to change the surface tension of the droplet produced in the source and the ion distribution. Further work is needed to fully understand the reasons. The following text has been added:
"The matrix effects identified in this study are likely due to the extracted samples being a complex mixture of different compounds, including a high proportion of inorganic ions that are extracted into water. This is likely to change the surface tension of the droplet produced in the ionisation source and the ion distribution. Further work is needed to fully understand the reasons."

Line 404, "Zero-dimensional box modelling indicates on some days up to 35 % of the isoprene-derived RO2 radicals can react with HO2 in the afternoon (Newland et al., 2019)." Please kindly note that this paper is under review or has been accepted for publication.

Newland et al., 2019 has now been published in ACPD. https://www.atmos-chem-phys-discuss.net/acp-2020-35/

Line 425, "Therefore, these spikes in 2-MT-OS could be a result of either higher 2-MTOS in regional aerosol transported to the site or a high isoprene emission source to the south west of the site (i.e. producing IEPOX locally) that then reacts with increased regional sulfate pollution." Any field evidences or model predictions show that the IEPOX can be effectively produced locally? What are the contributions from the regional transport? The effects of anthropogenic emission on iSOA formation observed in this work are local, regional or a combination of both effects.

We have added a sentence to clarify that the influence of anthropogenic pollutants on iSOA is occurring at both the local and regional level.

The following text has been added: ". The I-CIMS data shows that the IEPOX/ISOPOOH (Figure S5 and Newland et al., 2020) signal increases during the afternoon as the NO levels drop to below 1 ppb. The low NO levels mean that up to 30 % of the isoprene peroxy radical from OH oxidation can react with $HO_2$ rather than NO at this site, meaning IEPOX can be formed locally (Newland et al., 2020). There is also likely to be a regional source of IEPOX and 2-MT-OS, suggesting both local and regional

anthropogenic influences."

Line 450, "The ratio of 2-MT-OS:2-MG-OS observed in Beijing is compared to previous studies in Table 2 and is considerably lower than measurements taken in a range of isoprene dominated

environments (South East US, 2-MT-OS:2-MG-OS = 17, Budisolistiorini et al., 2015.; Amazon, 2-MTOS:2-MG-OS = 13-118, Glasius et al., (2018).; Atlanta, 2-MT-OS:2-MG-OS = 33, Hettiyadura et al., (2019)) reflecting the strong impact
of urban NO emission on iSOA formation." I agree with this argument. However, how could we use this ratio to quantify the effect of NO emission on iSOA formation or OS formation in different

regions?

We like the reviewers suggestion of using this kind of data as a metric for understanding the formation pathways in different environments. However, at present there is not enough information available for the ratio of these species in urban or suburban areas to develop this metric. This has been added as a

suggestion for future work.

The following text was added: "Future work will investigate how to use these ratios to quantify the effect of NO emission on iSOA formation in different regions."

Line 505, "Some of the NOS observed peaked in the daytime and some were enhanced at night. In total they had a mean concentration of 24 ng m-3 during the campaign. The sources and formation of these species will be discussed in a separate publication." I am okay with this. However, could the authors briefly elaborate how the detection of NOS help to better understand the effect of anthropogenic emission on the iSOA

formation in this work?
Thank you for your comment. We have added:

[revised manuscript text omitted]

---

## Author Response (AR2)

Reply to second round of reviewers comments

We would like to thank the reviewers for their further comments on the manuscript. Reviewers comments are in red, author replies are in Black and added text is in blue.

Report#1

Major

a) The authors did a good job in comparing the results from HILIC and RPLC, however, I was wondering why only the peak areas of 2-MT-OS were correlated. What about the 2-MG-OS?

2-MG-OS was not targeted during the HILIC analysis. This analysis was undertaken in a different lab, so unfortunately further experiments cannot be undertaken.

b) I disagree with the conclusion that "there is no evidence of ion source induced artifacts" (L527). As the authors acknowledge, there is still some co-elution of sulfate and nitrate ions with detected OS species. Therefore, there is still a chance of adduct formation between inorganic ions and organic compounds. The "test for artefact formation" conducted by the authors is not convincing, as it is just proving that sulfate and 2-methylglyceric acid / 2-methyltetrol do not produce artifacts. This cannot be generalized for all organic compounds! (By the way, I totally agree with the authors' conclusion that all previous direct infusion studies suffer from in-source adduct formation. Therefore, I'm actually very happy to see that people are now more and more using chromatographic separation before ionization. Nonetheless, if the separation is not done appropriately, we will not get away from ion source artifacts.)

The reviewer states there is a chance that adduct formation happens. However they provide no evidence to show this is possible. In contrast, we directly show that for the two main species we targeted in this study, artefacts do not form under our conditions. It is true that this does not mean all that all organics act the same way but we do not say that in the paper. With the nature of such complex samples (rough analysis shows over 5000 species in one ambient sample) co-elution is inevitable, even with the best chromatographic method. Throughout the entire chromatogram, not just during the first few minutes there is the co-elution of species, which would be the same for every ambient study. So, unless there is complete separation of all species in ambient samples, matrix effects are a possibility. However, we have at least been able to rule out the adduct formation from two authentic standards which to our knowledge the best has done at this stage of our understanding. In response to this comment we have added text to highlight that adduct formation COULD be occurring, but we have ruled out that organosulfate formation is not occurring from the co-elution with sulfate for these two authentic standards, and have said that more evidence is needed to show if adduct formation is occurring or not for other organic species.

The following text has been added:

L220 : "This therefore rules out adduct formation for the two most important iSOA species, 2-MT-OS and 2-MG-OS, however due to the lack of authentic standards and the complexity of the samples, adduct formation throughout the entire chromatogram could still be occurring. At this stage, there is not enough evidence to say either way if adducts are forming or not."

c) The selected settings of the Q Exactive mass spectrometer are really far from appropriate for a quantitative analysis approach. With the settings applied here (i.e., full scan at R=70,000 combined with ddMS2 Top 10), the authors obtain ~1 full scan spectrum per second (i.e, full scan + 10 x ddMS2

= 256 ms + 10 x 64 ms = 896 ms). As a rule of thumb, for accurate quantification, about 12 data points are needed to get a good peak quality in the extracted ion chromatograms. Therefore, the chromatographic peak width needed here would have been roughly 11–12 seconds, which is a quite broad peak in typical UHPLC applications and only reached with quite concentrated samples. In particular, since all the OS peaks elute quite early, I would be surprised to see that the authors obtained more than ~5–6 data points for each of the peaks here. In principle, it is possible to increase data quality by measuring replicates. However, the authors decided to measure each sample only once. Therefore, I think the concentrations reported here are connected to very large uncertainties (which could have been avoided easily).

Here we strong object to the comments of the reviewer. All peaks have ample data points, with the peaks being around 10-14s wide. We agree that the peaks are wide compared to optimum UHPLC conditions, however this method was developed not just to target isoprene organosulfates, but a wide range of compounds. As such, our concentrations are not connected to large uncertainties due to the method. From the calibration curve you can see that the linear fit is very good for both the external and standard addition calibration methods. Also the repeatability is very good, with a low standard error. If we were under sampling our peaks, we would not be able to obtain this sort of calibration curve.

d) Regarding the uncertainties mentioned above, it is still unclear to me how the authors estimate an uncertainty of 60% for the measured OS concentrations.

We have taken the average ratio of standard addition to the 2-MG-OS external calibration values given in table S1 and S2 to get a mean correction factor of 1.83. The uncertainty (60 %) is calculated as $2\sigma$ of the 6 values. This has been added to the text.

Following text has been added:

L456: "The uncertainty was calculated as $2\sigma$ of the 6 values used to calculate the average correction factor of 1.83."

Minor / Technical comments:
1) There are several mentions of "corplot" and "PollutionRose plot" in the manuscript, which might be disturbing to readers less familiar with R. I would recommend replacing these R-specific names with more general terms (e.g., corplot = correlation plot).

The terminology has been changed for the corplot, and a more detailed description added for the pollution rose plot.

Following text has been added:

L414: "in a correlation plot (R, Openair, CorPlot)"

L415: "The correlation plot"

Figure 5 caption: "Correlation plot (R, Openair, CorPlot)"

Figure 2 caption: "Figure 2. PollutionRose plot (Openair) of particulate sulfate measured by AMS, for the sampling period. Highlighting under what wind conditions the highest concentrations of sulfate occur."

2) Figure 5: Some of the label names capitalized and some not.

These have been capitalized.

[Figure]

3) L901: Ultra-performance liquid chromatography (UPLC) is actually specific for components from Waters Corporation. The more general term is ultra-high performance liquid chromatography (UHPLC).

This has now been changed throughout. On lines 45, 173, 175, 205, 206, 248, 426, 482, 530, 532, 659

4) Figure S3: I suppose the x-axis label should read 2-MG-OS.

This has now been changed.

[Figure]

$R^2 = 0.9965$

Report#2

Major:

Since AMS data is available in this campaign (L984), could the authors look into the tracer of IEPOX-SOA (i.e., m/z 82 or C_5H_6O^+) to see what is the relationship between the measured tracers and that ion tracer in AMS? Presumably NOx conditions may also affect the channels of IEPOX or OS formation if the chemistry is similar in other regions with SOA from isoprene.

The measured tracers (2-MT-OS and 2-MG-OS) showed limited correlation towards the C5H6O+ tracers with R2 values of 0.3837 and 0.4079 respectively. This is likely due to the low levels of IEPOX-SOA at this location and that a fraction of the C5H6O+ peak comes from other sources, such as methyl-furan. We do not feel this adds to the overall story of the paper and have not included it.

Minor:

1. The authors used "ultra-pressure liquid chromatography" in the Abstract, but "ultra-performance liquid chromatography" in L901. Please be consistent.

The terminology has now been changed to ultra-high performance liquid chromatography (UHPLC) throughout. On lines 45, 173, 175, 205, 206, 248, 426, 482, 530, 532, 659

2. Figure 7. Please use [O3] × [SO_4^2-] in the title of y axis. And better to use SO_4^2- for sulfate as there is no such thing as SO_4.

This has now been changed.